# BioAnalyst: A Foundation Model for Biodiversity

## Abstract

Multimodal Foundation Models (FMs) offer a path to learn general-purpose representations from heterogeneous ecological data, easily transferable to downstream tasks. However, practical biodiversity modelling remains fragmented; separate pipelines and models are built for each dataset and objective, which limits reuse across regions and taxa. In response, we present BioAnalyst, to our knowledge the first multimodal Foundation Model tailored to biodiversity analysis and conservation planning in Europe at $0.25°$ spatial resolution targeting regional to national-scale applications. BioAnalyst employs a transformer-based architecture, pre-trained on extensive multimodal datasets that align species occurrence records with remote sensing indicators, climate and environmental variables. Post pre-training, the model is adapted via lightweight roll-out fine-tuning to a range of downstream tasks, including joint species distribution modelling, biodiversity dynamics and population trend forecasting. The model is evaluated on two representative downstream use cases: (i) joint species distribution modelling and with 500 vascular plant species (ii) monthly climate linear probing with temperature and precipitation data. Our findings show that BioAnalyst can provide a strong baseline both for biotic and abiotic tasks, acting as a macroecological simulator with a yearly forecasting horizon and monthly resolution, offering the first application of this type of modelling in the biodiversity domain. We have open-sourced the model weights, training and fine-tuning pipelines to advance AI-driven ecological research.

## 1 Introduction

Biodiversity, encompassing the variety of all life forms on Earth, is fundamental to the stability and resilience of ecosystems. However, this rich diversity is under unprecedented threat due to numerous factors such as climate change (Gitay et al., 2003), pollution (Manisalidis et al., 2020; Azevedo-Santos et al., 2021), habitat destruction, over-exploitation of natural resources (Cordes et al., 2016), and the introduction of invasive species (Crystal-Ornelas & Lockwood, 2020). These pressures have led to significant declines in species populations and ecosystem degradation, posing critical risks to human well-being by compromising essential ecosystem services, such as clean air, water, and fertile soil (Cardinale et al., 2012).

Addressing these challenges requires predictive models to understand ecosystem dynamics and quantify the impacts of interventions. This, in turn, raises the overarching question of how to integrate such insights into decision-support frameworks for biodiversity conservation. Traditional methods often rely on static models, such as species distribution maps (Jung, 2023), which lack real-time updates and fail to capture rapid environmental changes. The fragmented nature of biodiversity data, dispersed across various sources and formats (Wohner et al., 2022), hinders effective data harmonisation and integration. Additionally, ecological systems are inherently complex and less understood compared to engineered systems, making accurate modelling and prediction arduous tasks. Uncertainties and knowledge gaps persist, particularly in identifying and accounting for unknown variables and intricate inter-species interactions (Zhu et al., 2022).

Recent advancements in AI and the development of Foundation Models offer promising avenues to overcome these challenges (Bommasani et al., 2021). FMs, pre-trained on large-scale datasets primarily through self-supervision, have revolutionised fields such as natural language processing

(Brown et al., 2020b) and computer vision (Dosovitskiy et al., 2020), demonstrating remarkable adaptability across diverse tasks. While geospatial foundation models are increasingly applied in ecological research, biodiversity modelling presents distinct challenges due to its reliance on unique data modalities, such as species occurrence records, trait databases, and fine-scale environmental covariates, for which specialised FMs have yet to be developed. The complexity and heterogeneity of ecological data, including species occurrence records, genetic sequences, remote sensing imagery, climate data, and environmental variables, pose significant challenges for integration and scalability. Moreover, the lack of standardised protocols for data collection and model development further complicates the creation of comprehensive AI tools in this domain (Trantas et al., 2023).

In response to these challenges, we introduce **BioAnalyst**, a Foundation Model specifically designed for biodiversity analytics, opening new avenues on both regional and national scale conservation planning efforts. Our contributions in this work are threefold:

- **Development of the first Multi-modal Biodiversity Foundation Model**: To our knowledge the first large-scale AI model tailored for biodiversity modelling, capable of processing and integrating diverse data types to model complex ecological phenomena.

- **Advancement in Predictive Biodiversity Analytics**: We demonstrate BioAnalyst's predictive capabilities in key applications such as species distribution modelling, biotic and abiotic reconstruction, and population trend forecasting, especially in data-scarce scenarios.

- **Open Collaboration and Resource Sharing**: By openly releasing BioAnalyst's code, weights, and fine-tuning workflows, we aim to foster collaborative efforts within the scientific community, thereby accelerating research and conservation initiatives that address pressing ecological challenges.

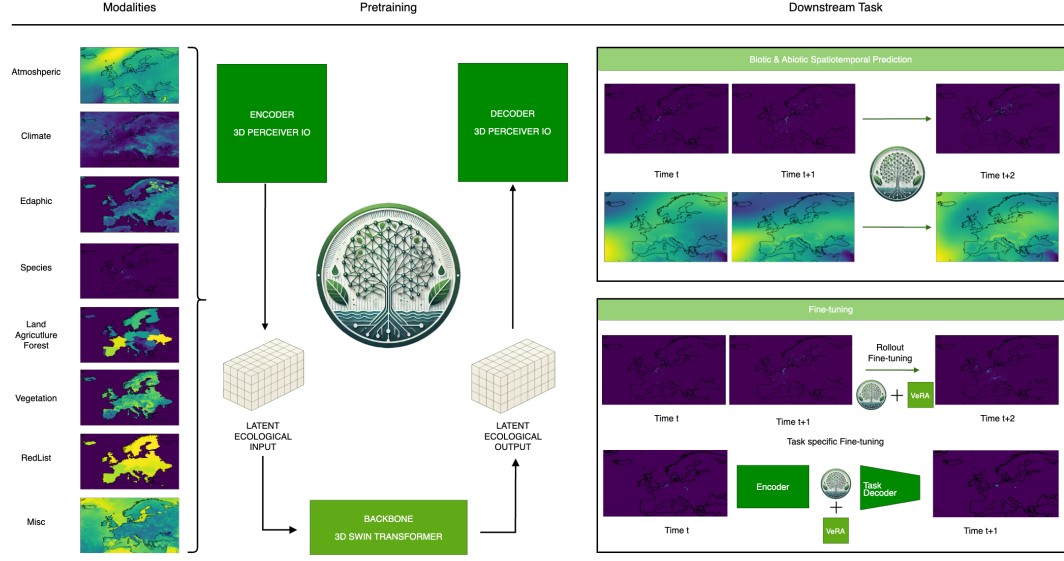

Figure 1: BioAnalyst is the first large-scale multi-modal model for biodiversity, trained on 20 years of spatiotemporal data modalities. The model ingests 10 distinct modalities, encoding and aligning them to latent ecological representations via the 3D Perceiver IO encoder. It then processes the latent space with the 3D Swin Transformer backbone and decodes it back to produce accurate spatiotemporal predictions. BioAnalyst shows strong performance in downstream tasks like (i)biotic, (ii) abiotic features prediction, (iii) long horizon prediction (12 timesteps = 1 year), both across space and time and (iv) is easily fine-tunable for any downstream task.

## 2 RELATED WORK

One of the first successful applications of FMs in Earth Sciences involves a geospatial foundation model trained on raw satellite imagery, called *Prithvi*, which can tackle tasks such as flood mapping, wildfire scar segmentation, multi-temporal crop segmentation, and cloud gap imputation (Jakubik et al., 2023). A follow-up work introduced *Prithvi WxC*, a larger 2.2 billion-parameter FM that emulates weather and climate phenomena in tasks such as autoregressive rollout forecasting, downscaling, gravity wave parameterisation, and extreme events estimation (Schmude et al., 2024). On the same theme, *Pangu-Weather* delivered higher performance in medium-range forecasting, improving numeric weather prediction methods by training a 3D transformer model on 39 years of global data, which injects Earth-specific priors (Bi et al., 2023).

Focusing on Earth system dynamics and predictability, as well as the more specific and accurate prediction of extreme weather and climate events, *ORBIT* showcased advanced performance and highlighted the requirement for High Performance Computing (HPC) (Wang et al., 2024). Similarly, *Aurora* can produce operational forecasts for global medium-range weather with unprecedented accuracy and speed-up over classical numerical weather prediction (NWP) models, by combining a flexible 3D Transformer backbone with a distinct encoder-decoder architecture (Bodnar et al., 2024). Following similar approaches, *Aardvak Weather* features an end-to-end pipeline for data-driven weather prediction focusing on computation and maintenance benefits compared to classic NWP models (Allen et al., 2025). In the Earth Observation domain, *TerraMind* is a large FM pre-trained on nine distinct modalities, highlighting the powerful alignment of token and pixel-level representations. In addition, this work demonstrates both the benefits of early fusion on downstream tasks and the performance gains achieved when learning on modalities generated by the FM (Jakubik et al., 2025). To stimulate the development of FMs for earth monitoring, *GEO-bench* offers a suite of six classification and six segmentation tasks, suited for model evaluation (Lacoste et al., 2024).

In ecology, the number of FMs is relatively small, with a focus on visual, audio, and natural language tasks. More specifically, *BioCLIP* is an FM classifier for biology for the tree of life, trained on the TREEOFLIFE-10M, namely the abundance and variety of images of plants, animals, and fungi, together with the availability of rich structured biological knowledge (Stevens et al., 2024). Similarly, *Insect-Foundation* introduced a 1M dataset with insect imagery and taxonomy, and an FM based on ViT backbone (Dosovitskiy et al., 2020) trained on this dataset for classification (Nguyen et al., 2024). For species distribution modelling, *NicheFlow* demonstrated good predictive performance, mainly in reptile species (Dinnage, 2024), employing a Variational Autoencoder architecture and using environmental and species distribution variables. Combining audio and textual information *NatureLM* uses a pretrained encoder and a frozen LLM backbone (Llama 3.1-8b) to produce a text sequence used for bioacoustics tasks and more specific species-classification and detection (Robinson et al., 2024).

## 3 METHOD

BioAnalyst has been designed to utilise the predictive power of the latest AI transformer-based models while being flexible enough to digest multi-modal geospatial input variables. Our work is inspired by the development of large-scale climate and weather models, such as Aurora (Bodnar et al., 2024) and Prithvi (Schmude et al., 2024; Jakubik et al., 2023), extending the capabilities of Foundation Models in the domain of biodiversity. More specifically, we are interested in learning about and forecasting biodiversity dynamics at both regional and national scales with adequate resolution.

The design choices of BioAnalyst were driven by specific capabilities that should possess, including multi-modal data representation, spatiotemporal feature preservation, regional and national operation, underlying physics simulation across multiple scales, and various use-case adaptability. We selected 28 km resolution for BioAnalyst, as it is appropriate for regional to national-scale applications such as (i) mapping broad richness and community-composition patterns, (ii) identifying large-scale hotspots and coldspots under different climate scenarios, and (iii) informing high-level prioritization or reporting (e.g. national assessments, EU-wide strategies). To account for them, BioAnalyst can be thought of as a *forecast emulator*, i.e., given a state of the Earth's biodiversity

at times $t$ and $t - \delta t$, it predicts the state at $t + \delta t$, where $\delta$ is a discrete time step and the final fine-tuned model can predict up to 12 time-steps ahead. Although this might seem very simple, it poses significant challenges for modelling and engineering in complex domains such as ecology and, more specifically, biodiversity, which we have attempted to tackle to the best of our ability given the constrained resources at hand.

## 3.1 FOUNDATION MODEL ARCHITECTURE

Forecasting is a common task in Earth Sciences, such as weather, climate, and ecology, which is mainly modelled with time-series methods. Related work on weather and climate utilises the latest advancements in computer vision literature, including masked autoencoders (He et al., 2022), which exploit their low memory footprint, masking properties, and the handling of ungridded and sparse observation data.

BioAnalyst implements an encoder-backbone-decoder architecture. Let the input data at some time $t$ be a multi-modal tensors $\mathbf{X}_t \in \mathbb{R}^{\mathcal{H} \times \mathcal{W} \times C_{in}}$, representing $C_{in}$ variables over a spatial grid of height $\mathcal{H}$ and width $\mathcal{W}$. The model components are:

- **Encoder** $\mathcal{E}$**:** We use Perceiver IO (Jaegle et al., 2021b;a), a general-purpose attention architecture. Input variables $\mathbf{X}_t$ are first tokenized into $N_p = \mathcal{H}/p \times \mathcal{W}/p$ non-overlapping patches of size $p \times p$. Fourier features encode spatial coordinates, which are combined with learned embeddings for variable types, time steps, and atmospheric levels. The resulting features associated with each patch are projected into the model's embedding dimension $D_e$, creating tokens $\mathbf{T}_t \in \mathbb{R}^{N_p \times D_e}$. These are processed by the Perceiver IO encoder $\mathcal{E}$, which maps them to a fixed-size latent array $\mathbf{Z}_t \in \mathbb{R}^{N_l \times D_e}$ (where $N_l$ is the number of latent tokens) using cross-attention followed by self-attention layers:

$$\mathbf{Z}_t = \mathcal{E}(\mathbf{T}_t) \tag{1}$$

- **Backbone** $\mathcal{B}$**:** We use a SwinTransformer (Liu et al., 2021) as the neural simulation engine. It receives the latent representations from two previous steps $\mathbf{Z}_{t-1}, \mathbf{Z}_t$ and predicts the next latent state $\mathbf{Z}'_{t+1}$ using hierarchical stages with shifted window self-attention:

$$\mathbf{Z}'_{t+1} = \mathcal{B}(\mathbf{Z}_{t-1}, \mathbf{Z}_t) \tag{2}$$

This part aims to enable efficient computation while capturing spatial dependencies at various scales, thereby emulating the system dynamics in the latent space.

- **Decoder** $\mathcal{D}$**:** The same Perceiver IO model is used to reconstruct the output variables. It makes use of specific query tensors $\mathbf{Q} \in \mathbb{R}^{N_q \times D_e}$ corresponding to the desired output variables (total $N_q$ features) and their coordinates on the target grid. These queries attend to the backbone's output latent state $\mathbf{Z}'_{t+1}$ via cross-attention within the decoder $\mathcal{D}$. The decoder outputs a sequence $\hat{\mathbf{Y}}_{t+1} \in \mathbb{R}^{N_q \times D_e}$ which is then projected and reshaped to the final multi-modal feature grid $\hat{\mathbf{X}}_{t+1} \in \mathbb{R}^{\mathcal{H} \times \mathcal{W} \times \mathbf{C_{out}}}$ such that:

$$\hat{\mathbf{Y}}_{t+1} = \mathcal{D}(\mathbf{Z}'_{t+1}, \mathbf{Q}) \quad \xrightarrow{\text{Reshape}} \quad \hat{\mathbf{X}}_{t+1} \tag{3}$$

The design choices for BioAnalyst prioritise learning informative features in a compact latent representation before the emulation stage. By using Perceiver IO for both encoding and decoding stages, we aim to learn features from the original (raw) input data, thereby avoiding the standard approach of using separate tokenisation pipelines for each modality/variable-type, which can lead to biased tokens that are heavily dependent on the model used to produce them. This unified approach enables the model to capture cross-modal interactions at different data granularity levels, allowing it to differentiate features across various domains, from ground conditions and atmospheric levels to species distributions. More detailed information on the architecture is available in Appendix A.

## 3.2 PRE-TRAINING DATA SELECTION

BioAnalyst is pre-trained on BioCube (Stasinos et al., 2025), which compiles and aligns multiple datasets into a fixed spatio-temporal cube. Our main driver is modelling biodiversity "as-a-whole", which means we require observations from below the surface, the surface and above. Our analysis

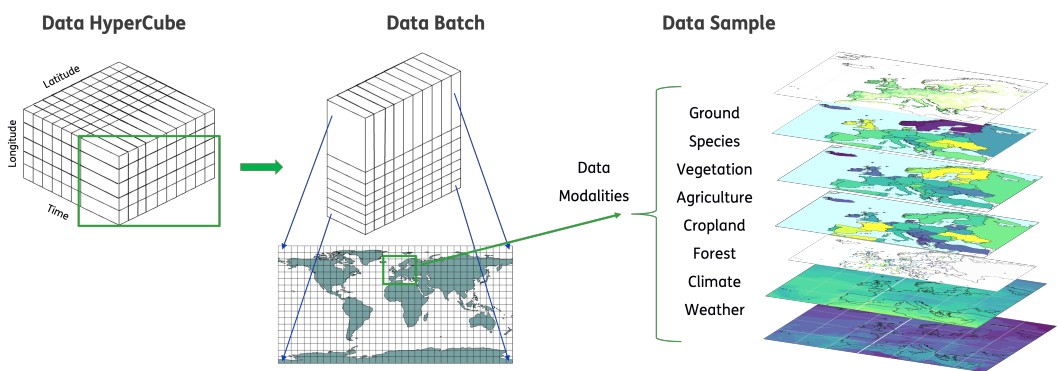

Figure 2: A visual explanation of the data pipeline. From left to right, we received the data from BioCube in a HyperCube format, where sampling a single-timestep slice produces a Data Batch containing worldwide observations. Selecting European coordinates produces a Data Sample with multiple modalities stacked on the selected coordinate grid of size [160, 280].

is confined to terrestrial (land-based) biodiversity; therefore, datasets describing marine or coastal biota are intentionally excluded from the study. Additionally, we select a subset of the total available features, categorised by variable groups, namely: surface variables, atmospheric variables with 13 levels, climate variables, edaphic variables, vegetation variables, species distribution variables, land, agriculture and forest variables, redlist variable and miscellaneous variables. Our species distribution variables consist of animal species only and not plants.

These features are combined in a Data HyperCube, grounded on the coordinate reference system WGS84. The HyperCube contains global coordinates with a resolution of 0.25 degrees (grid sampling $\sim$ 28 km) from the whole world while our focus is on European biodiversity, leading us to select a slice from it, yielding a Data Batch from [latitude: $\mathcal{H}$, longitude: $\mathcal{W}$ ] = [(32, 72), (-25, 45)] = [160, 280]. The observation time range spans from January 1, 2000, to June 1, 2020, and we sample with a 1-month lead time from this range, resulting in total to 233 monthly Data Samples.

Selecting a Data Sample from the Data Batch yields a composite multi-modal cell of European coordinates, with a specific monthly time-stamp. Each of these Data Points for $t \in [0, 233]$ contains 124 observations per location cell. The total observations are calculated by summing the 11 variable groups with the number of variables they contain and for the atmospheric variables multiplying the variables with the number of pressure levels and adding them to the sum. More specifically, we denote the observed data points at a discrete time $t$ by a collection of tensors $\mathbf{X}_t$:

$$\mathbf{X}_t \in \mathbb{R}^{\mathcal{H} \times \mathcal{W} \times F}, \tag{4}$$

with $\mathcal{H} = 160$, $\mathcal{W} = 280$, and $F = 124$ feature channels per spatial cell. We consider $I = 11$ variable groups indexed by $i = 1, \dots, I$, where the $i$-th group at time $t$ is represented as

$$V_t^{(i)} \in \mathbb{R}^{\mathcal{H} \times \mathcal{W} \times L_i}, \tag{5}$$

with $L_i$ the number of levels associated with that group (e.g. 13 pressure levels for atmospheric variables or a single level for all other variable groups). A full Data Sample is obtained by concatenating all variable-group tensors along the feature dimension,

$$\mathbf{X}_t = \text{concat}_{i=1}^{I} V_t^{(i)}, \tag{6}$$

so that each spatial cell $(\mathcal{H}, \mathcal{W})$ at time $t$ is associated with a feature vector $x_{t,h,w}^v \in \mathbb{R}^{124}, v \in V$.

During pre-training and roll-out fine-tuning, the input data consist of tuples of two consecutive time-steps. A generic input at time index $t$ is given by

$$\left(\mathbf{X}_t, \mathbf{X}_{t+1}\right) \in \left(\mathbb{R}^{\mathcal{H} \times \mathcal{W} \times F}\right)^2, \tag{7}$$

A visual representation of the above can be found on Figure 2 while the complete data description is available in Appendix B.

### 3.3 Pre-training Objective

Pre-training climate and weather FMs for forecasting is frequently done by minimising a performance metric, either Mean Squared Error (MSE) or Mean Absolute Error (MAE). A straightforward approach is to force the model's output at time $t + 1$ to match the known future data. Formally, for a single variable $v$:

$$\mathcal{L}_{MAE} = ||\hat{x}_{t+1}^v - x_{t+1}^v|| \tag{8}$$

where $\hat{x}_{t+1}^v$ is the model's prediction for a variable $v$ at time $t + 1$. Summing across all variables and levels provides a multi-target objective.

In ecological contexts, temporal difference learning can be beneficial. Instead of predicting $x_{t+1}$ directly, we predict the increment $\Delta x_t = x_{t+1} - x_t$. This approach, often encountered in reinforcement learning (Sutton & Barto, 2018), can reduce biases from unobserved global offsets or stable large-scale means. For instance, daily vegetation changes or seasonal fluctuations in species population can be smaller and more stable than absolute population numbers. By focusing on differences, we encourage the model to learn transition dynamics or, more specifically, **biodiversity dynamics**:

$$\mathcal{L}_{TD} = ||\hat{\Delta x_t^v} - (x_{t+1}^v - x_t^v)|| \tag{9}$$

This choice is reinforced by empirical results in specific biodiversity modelling tasks (e.g., ephemeral wetlands or short-lived insect populations), which show improved forecasting stability over standard next-step MSE (Clark et al., 2001).

### 3.4 Fine-tuning

BioAnalyst is fine-tuned in two different settings, each contributing to a distinct goal. In this section, a description of each setting is provided, and in the next section, we present the quantitative results.

**Roll-out finetuning**: In this setting, we fine-tune the entire BioAnalyst for six and twelve rollout steps, effectively predicting biodiversity dynamics six months and one year ahead. Fine-tuning with a mix of variable horizons is a technique successfully used on Aurora (Bodnar et al., 2024), Prithvi-WxC (Schmude et al., 2024) and Aardvak (Allen et al., 2025), enhancing long-horizon forecasting capabilities. In practice, we freeze the whole architecture, including the encoder, decoder, and backbone, while training only the newly added VeRA adapters (Kopiczko et al., 2024) on the backbone's attention heads. We found VeRA to perform equally or sometimes slightly better than other Parameter Efficient Fine-tuning Techniques (PEFTs), such as LoRA (Hu et al., 2022), which use only one-tenth of the learnable parameters. More specific, starting from an observed state $x_t$, the model is unrolled autoregressively: at each step $k$ it takes the previous prediction $\hat{x}_{t+k-1}$ as input and produces a new prediction $\hat{x}_{t+k}$. We supervise all intermediate steps with the same task-dependent loss $\mathcal{L}_{TD}$ and define the $K$-step rollout loss as

$$\mathcal{L}_{\text{rollout}}(t, K) = \frac{1}{K} \sum_{k=1}^{K} \mathcal{L}_{TD}\left(\hat{x}_{t+k}^u, x_{t+k}^u\right). \tag{10}$$

Our mix horizons are $K = 6$ and $K = 12$ months. To keep training stable and affordable for long sequences, we follow the "pushforward trick" of Brandstetter et al. (2022): during backpropagation we stop gradients at the inputs of steps $k > 1$, so that the loss is averaged over all steps for evaluation, but only the final step ($k = K$) contributes gradients to the model parameters.

**Task specific finetuning**: To evaluate the ecological capacity and environmental structure encoded in BioAnalyst, we implement two complementary fine-tuning tasks. These tasks are designed to interrogate distinct dimensions of the model's representation space: its ability to adapt to biotic presence-only data under temporal shift, and its capacity to retain structured abiotic gradients related to seasonal climate. For comparison, we also benchmark the performance on these fine-tuning tasks using four baselines and the Aurora-0.25 model (Bodnar et al., 2024), which shares the same model architecture but has been pre-trained on climate modalities only. More specific, the first is a joint species distribution task that involves partial model adaptation: the BioAnalyst's encoder and

decoder are frozen. At the same time, the backbone is fine-tuned with VeRA adapters using historical (2017-2021) species vascular plant occurrence (500 most commonly occurring species) data from the GeoLifeCLEF2024 benchmark dataset (Joly et al., 2024) to forecast time-series distributions. The second task is diagnostic: a regression head is trained on top of frozen decoder embeddings for both BioAnalyst and Aurora, to predict monthly climate variables (2-meter temperature and precipitation) from CHELSA v2.1 (Karger et al., 2017) (2000-2019). Together, these tasks provide a dual lens on the model's ecological generalisation and environmental coherence. The detailed information for the model card, software, training and scaling, finetuning tasks and metrics is available at Appendix C.

## 4 RESULTS

### 4.1 ROLLOUTS: FORECASTING BIODIVERSITY DYNAMICS

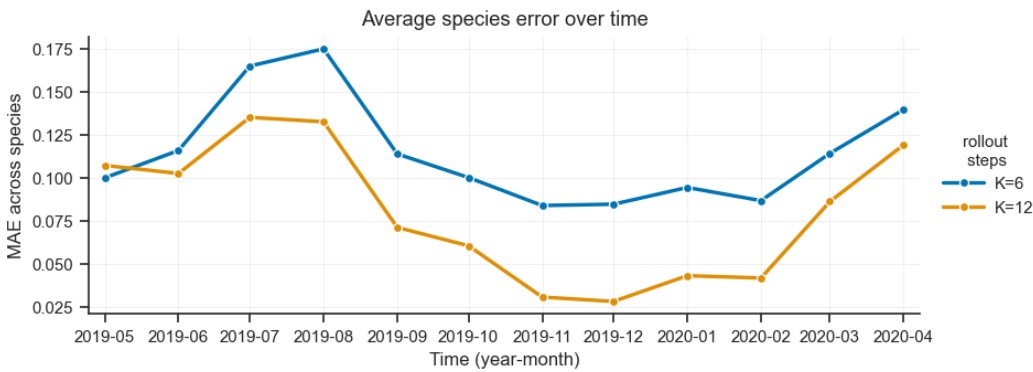

Figure 3: Mean Absolute Error for the 28 animal species on a 12-step rollout. Blue line highlights the performance of roll-out finetuned BioAnalyst for $K = 6$ steps while orange line for $K = 12$ steps.

In this part, we present four main results produced from our pre-training and rollout fine-tuning experiments whose target is to model and forecast biodiversity dynamics. First we present the MAE for the species variable group (28 animal species), comparing the performance of a 12-step roll-out for our mix horizons $K = 6$ and $K = 12$ time-steps respectively Figure 3. The 12-step finetuned variant follows a lower MAE across the 12-step trajectory, validating the effectiveness of our mix horizon scheme during roll-out finetuning. Both variants exhibit increasing MAE after 10 steps. Second, we report a mean Sørensen similarity of 0.31 for 28 animal species across all evaluated land grid cells, indicating that the predicted assemblages recover roughly one-third of the observed community composition. Figure 4 highlights spatial variation in assemblage agreement, with higher similarity in western and central Europe and lower values in eastern and south-eastern regions. This pattern closely follows known spatial biases in the underlying GBIF occurrence data, with denser sampling in some regions than others, so the map reflects both model skill and uneven observation effort. Overall, the spatial pattern suggests moderate compositional skill in well-sampled areas, useful for broad biogeographic inference, yet leaving room for improving species co-occurrence dynamics. Third, Figure 5 depicts BioAnalyst ability to localise and spatially predict granular species distributions. Fourth, our investigation on BioAnalysts cross-attention weights, highlighted that climate variables receive the highest attention, followed by the species and surface variables, while most other modality groups receive lower attention. Spatial analysis, particularly in the case of a high-interest variable (i.e., species), reveals that attention is not uniformly distributed, rather than concentrates on Scandinavia and Central Europe regions Appendix D.

### 4.2 BIOTIC FINE-TUNING: FORECASTING SPECIES DISTRIBUTIONS

To assess the biotic predictive capacity of BioAnalyst's pre-trained embeddings, we fine-tuned a classification head to forecast species presence-absence for 500 vascular plant species in the GeoLifeCLEF 2021 dataset. The model achieved high accuracy with an F1 score of 0.9964 and an

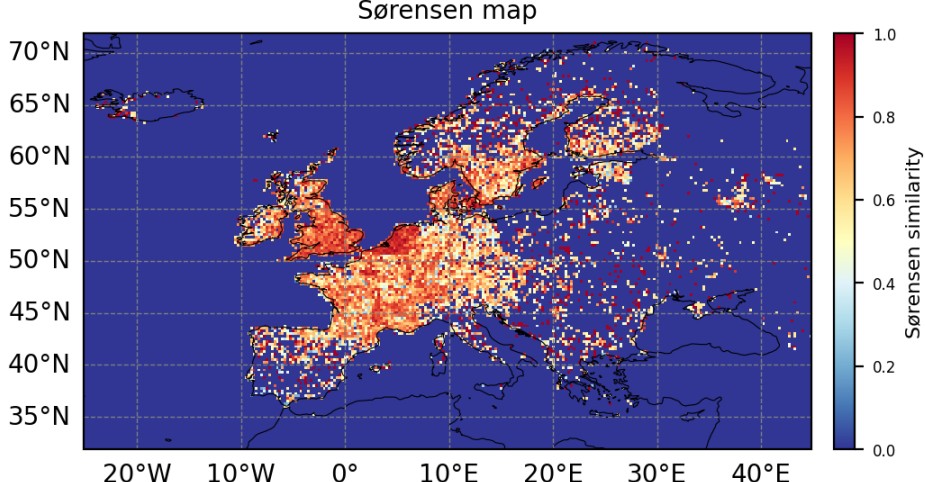

Figure 4: Community Sørensen similarity between predicted and observed plant assemblages for 28 animal species across Europe, based on GBIF presence records. The mean similarity is $\bar{S} = 0.31$, indicating that the model recovers roughly one third of the recorded community composition. Warm colours (yellow–red) denote higher assemblage agreement, while cool blues indicate little overlap and/or sparsely sampled regions.

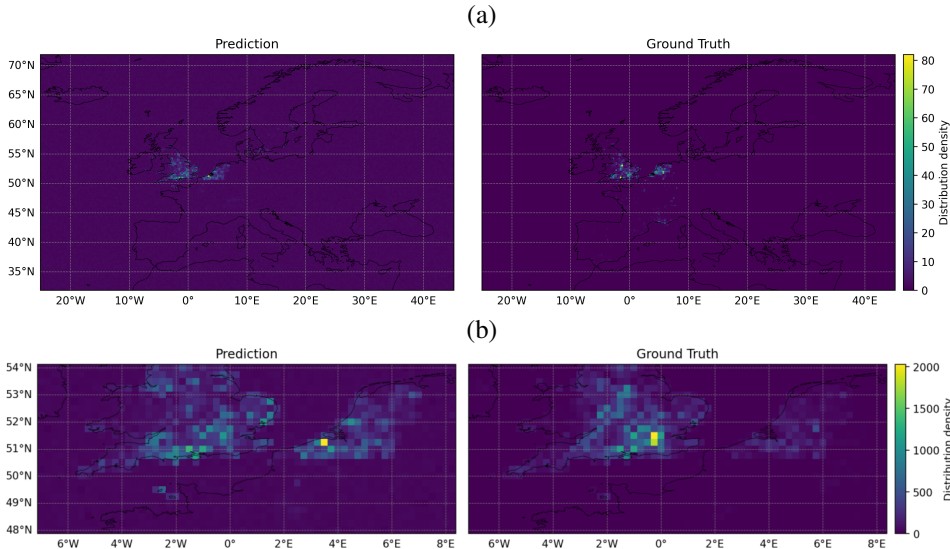

Figure 5: (a) Ground truth and prediction spatial plots for the species *Pieris brassicae* (ID 1920506) on 01-4-2019 and MAE of 0.00003. (b) Zoomed in plot, highlighting areas of interest, where the model can capture the general distribution, although unable to capture high-density areas.

RMSE of 0.5284, surpassing our selected baselines LatentMLP and ConvLST, demonstrating its ability to support macro-scale joint species distribution modelling. In the same setting, Aurora performs marginally worse in evaluation metrics than BioAnalyst, as shown in Table 1. Nevertheless, investigating the predicted species richness in more detail for both models, we observe that Aurora produces a more spatially extensive richness field, assigning moderate richness also to cells with recorded absent observations, which likely reflects extrapolation from smooth climate fields. Bio-Analyst's predictions remain more tightly constrained to the observed footprint, consistent with the additional process-related modalities it uses (land use, vegetation, species signals) Figure 6. This result supports the view that, for complex community-level processes, climate-only representations

can be limiting, and that integrating additional process-related modalities, as in BioAnalyst, can substantially improve predictive performance and spatial realism.

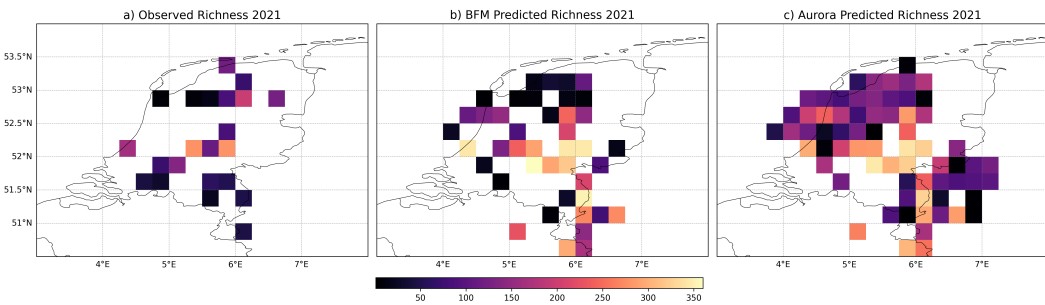

Figure 6: Comparison of observed and predicted species richness in the Netherlands for the year 2021. a) The Observed Richness based on empirical field data from GeoLifeCLEF2024 presence-absence survey; blank cells represent true zeros indicating absence of species occurrence. b) The species richness for 2021 predicted by the BioAnalyst, and c) Aurora-0.25 predictions for 2021.

Table 1: Performance of the species distribution forecasting and of the linear probing task. Note: $R^2$ is not applicable to this classification-style species forecasting task and therefore not reported.

| Model | species distribution forecast | | | linear probing | | |
|---|---|---|---|---|---|---|
| | Loss | F1 | RMSE | Loss | $R^2$ | RMSE |
| BioAnalyst | 0.0057 | 0.9964 | 0.5284 | 0.0225 | 0.9002 | 0.1499 |
| Aurora | 0.0130 | 0.9945 | 0.5014 | 0.2668 | 0.7354 | 0.5144 |
| RandomForest | – | – | – | – | 0.7260 | 0.2426 |
| SVM | – | – | – | – | 0.1741 | 0.3828 |
| LatentMLP | 0.0435 | 0.8916 | 0.6951 | – | – | – |
| ConvLSTM | 0.0256 | 0.9924 | 0.5433 | – | – | – |

### 4.3 ABIOTIC LINEAR PROBING: RECOVERING SEASONAL CLIMATE STRUCTURE

The model achieved strong predictive performance, with the best epoch reaching an $R^2$ of 0.9002, a loss of 0.0225, and an RMSE of 0.1499, as shown in Table 1. These values indicate that the Bio-Analyst decoder outputs contain sufficient information to reconstruct fine-grained seasonal climate patterns even without any fine-tuning of the encoder Figure 7. The decoded precipitation captures major spatial gradients and orographic patterns (e.g., Alps, Norway), though with some smoothing. The decoded temperature field accurately reproduces latitudinal and coastal gradients present in the target. In the same theme, comparing BioAnalyst with baselines such as the pre-trained Aurora-025, a Random Forest model and Support Vector Machine yielded a stronger predictive capacity for Bio-Analyst. However, this comparison should be interpreted only as an indicative baseline: BioAnalyst is pre-trained directly on monthly biodiversity states, whereas Aurora is an Earth-system foundation model that is pre-trained to forecast geophysical variables at short lead times (typically 6-hour steps) and fine-tuned for weather, wave and air-quality tasks rather than biodiversity Appendix C.4.4.

## 5 CONCLUSION & DISCUSSION

In this work, we introduced BioAnalyst, to our knowledge the first Foundation Model for biodiversity. BioAnalyst is truly multi-modal, light-weight and can be used to model various complex spatio-temporal ecological phenomena with competitive performance for regional and nation-scale applications at $0.25°$ resolution, setting a new accuracy benchmark for ecological forecasting. We highlighted its predictive analytics as a stand-alone model in tasks such as biodiversity dynamics modelling, as well as in tasks like joint species distribution forecasting, absence detection, and monthly climate linear probing. This opens new opportunities for rapid model adaptation in

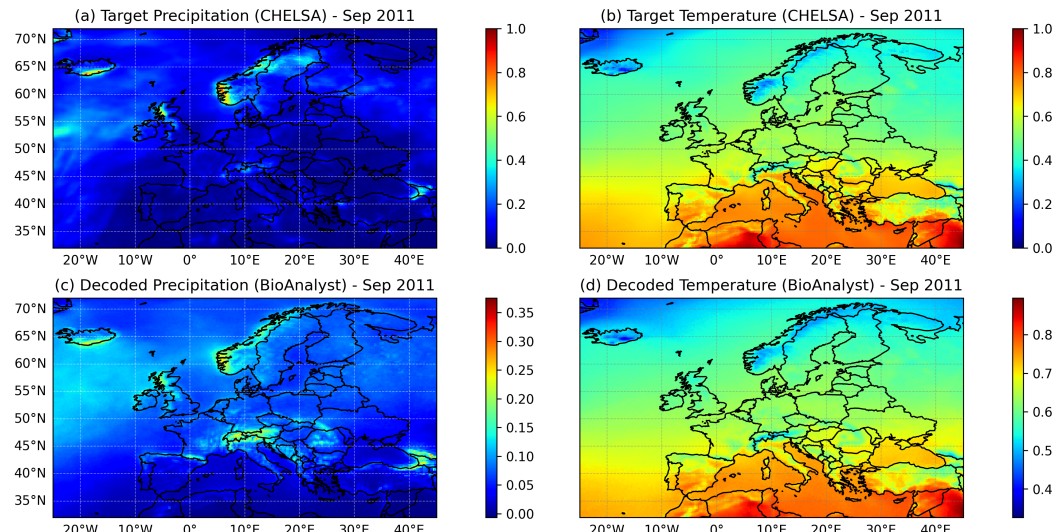

Figure 7: Comparison between CHELSA target climate fields and decoded outputs from the BioAnalyst decoder for September 2011 over Europe. Panels (a) and (b) show the downsampled CHELSA reference for precipitation (kg $m^2$) and temperature (K/10), respectively. Panels (c) and (d) show the corresponding decoded predictions from the model after reconstruction.

data-poor contexts and for advancing hypothesis-driven ecological modelling through representation learning.

However, some caution is warranted when interpreting downstream predictions. For instance, while BioAnalyst captures species occurrence and absence patterns with adequate accuracy (Figure 6), this success may partly reflect temporal biases in observation effort rather than actual ecological change. Although this modelling setup focuses on presence–absence, the multi-taxon nature of the dataset effectively means we are performing joint species distribution modelling (jSDM) across both space and time, which is an notoriously difficult task in classical ecology Tikhonov et al. (2020); König et al. (2021). Also, BioAnalyst, comes with a series of limitations like the lack of uncertainty quantification of the predictions, the constrained area of operation, Europe, which is not representative of global biodiversity dynamics. Since BioAnalyst is trained on data that represent biodiversity only in the terrestrial realm, it does not take into consideration the seas/oceans/lakes/dykes, for example, amplifying the bias towards other parts of biodiversity and the compound effects they may have on both regional and national biodiversity dynamics. Applications like local habitat management and single-site reserve design will require downscaling or complementary, higher-resolution data for further fine-tuning. Still, there are many open challenges ahead, one of the most difficult to tackle is data availability and methods to improve ecology-based downstream tasks.

Part of the future work will include more frequently sampled, higher resolution and better-curated data points that can further enhance BioAnalyst's capabilities in combination with longer training. A user interaction interface could enhance BioAnalyst's interpretability and ease of use. Methods for quantifying this uncertainty at different granularity levels, such as cartograms (Rocchini et al., 2019) or meta-model traits (Okánik et al., 2024), are planned as BioAnalyst's future work. Truly synergistic models that embed ecological principles (e.g. energy budgets, trophic interactions) into AI architectures that simulate population dynamics are an exciting frontier (Agarwal et al., 2025). This could mean neural networks that respect mass-balance constraints or reinforcement learning agents that simulate animal foraging behaviour. Such integration would yield models that not only predict well but also adhere to known ecological laws, making them more generalisable, trustworthy, and aligned with global biodiversity goals (DeSantis et al., 2025).

REPRODUCIBILITY STATEMENT

The pre-training and rollout fine-tuning codebase is available at `https://github.com/confsubmission26/bfm-model`. The task specific fine-tuning codebase is available at `https://github.com/confsubmission26/bfm-finetune`. For BioAnalyst implementation details, architecture blueprint and design choices, please look at Appendix A. For the complete pre-training and task fine-tuning data information, please look at Appendix B. The model card, pre-training and fine-tuning configurations are available at Appendix C. Additional ablation studies and interpretability material are available at Appendix D

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

APPENDIX

## A  BIOANALYST IMPLEMENTATION DETAILS

This appendix provides a detailed description of the BioAnalyst model's architecture, as a supplement to section 3.1.

### A.1  CORE ARCHITECTURAL BLUEPRINT

The BioAnalyst model is structured as an encoder-backbone-decoder system. Let the input data at time $t$ be a multi-modal tensor $\mathbf{X}_t \in \mathbb{R}^{\mathcal{H} \times \mathcal{W} \times C_{in}}$, where $\mathcal{H}$ and $\mathcal{W}$ represent the spatial height and width of the input grid, and $C_{in}$ denotes the number of input variables (channels).

Firstly, $\mathbf{X}_t$ is first processed by an **encoder** $\mathcal{E}$ module built upon Perceiver IO, which transforms the input data into a fixed-size latent representation $\mathbf{Z}_t \in \mathbb{R}^{N_l \times D_e}$, where $N_l$ is the number of latent tokens and $D_e$ is the embedding dimensions. The encoder can process inputs from one or more time steps (e.g., $t$ and $t-1$) to form $\mathbf{Z}_t$. This stage includes methods, including positional and temporal encodings, as detailed in later sub-appendices.

Next, $\mathbf{Z}_t$ is then fed into a **backbone** network $\mathcal{B}$. BioAnalyst muses a Swin Transformer as its backbone. The Swin Transformer processes $\mathbf{Z}_t$ through hierarchical stages with shifted window self-attention to model spatio-temporal dynamics and predict $\mathbf{Z}'_{t+1}$.

Finally, $\mathbf{Z}'_{t+1}$ is passed to a **decoder** $\mathcal{D}$, which uses a set of learnable query vectors $\mathbf{Q}$ corresponding to the desired output variables and their target grid locations, which attend to $\mathbf{Z}'_{t+1}$ to reconstruct the multi-modal feature grid $\hat{\mathbf{X}}_{t+1} \in \mathbb{R}^{\mathcal{H} \times \mathcal{W} \times C_{out}}$.

### A.2  DESIGN CHOICES

We have chosen Swin Transformer as our backbone because biodiversity patterns can be observed at multiple spatial scales (e.g., from microclimates to continental zones), and Swin's hierarchical architecture with patch merging/splitting appeared as an interesting and suitable choice for a backbone that would ingest data with multi-scale structuring. Additionally, its window-based attention mechanism also is stated as having linear complexity (when compared to the original ViT) in processing image/image-like inputs, making it a computationally efficient choice - a property that certainly would be desirable in the case of processing/training/testing on large inputs (e.g., input data spanning our entire planet, not just continental Europe). Additionally, we adjusted the architecture to support temporal conditioning through lead-time embeddings that can modulate processing at each stage, allowing the model to adapt representations based on the prediction time horizon, which is a capability we use for multi-month forecasting.

Integrating Perceiver IO as encoder-decoder and Swin Transformer as backbone is achieved through a simple reshape operation that reinterprets the latent sequence as a 3D spatial volume. The Perceiver IO encoder outputs a flat sequence of latent tokens, which is then reshaped into a spatial tensor with dimensions (depth, height, width), where the depth dimensions naturally aligns with the number of modality groups and the spatial dimensions correspond to the patch grid. This creates a multi-channel feature map, similarly to standard computer vision inputs, which the Swin backbone can then process through hierarchical window-based attention on the spatial dimensions while treating depth as feature channels. After Swin processing, the inverse transformation flattens the spatial volume back into a sequential format for the Perceiver IO decoder, which uses cross-attention to generate pixel-level predictions. This design combines Perceiver IO's strength at flexible, modality-agnostic encoding with Swin's hierarchical processing. The depth alignment with modality groups is by design: the encoder initialises separate latent parameter sets for each modality group (each containing one latent per spatial patch), which are concatenated in a structured order before Perceiver IO processing, which ensures that the subsequent, deterministic reshaping will map each modality to its own depth slice in the 3D volume.

### A.3  THE BIOANALYST ENCODER

The encoder $\mathcal{E}$ transforms the raw, multi-modal input tensor $\mathbf{X}_t$ into a structured, fixed-size latent representation $\mathbf{Z}_t$.

### A.3.1 INPUT PROCESSING AND FEATURE ENGINEERING

The spatial dimensions ($\mathcal{H}$ and $\mathcal{W}$) of each variable in $\mathbf{X}_t$ are first divided into non-overlapping patches. Each patch is of size $p \times p$ (where we kept $p = 4$ in BioAnalyst's configuration). This results in $N_p = (\mathcal{H}/p) \times (\mathcal{W}/p)$ patches per variable. For each variable type, the data within each patch potentially spanning multiple channels (e.g., different variables within a group), is flattened and then linearly projected to form initial patch tokens. To provide rich contextual information, these tokens are combined with several learned embeddings:

- **Spatial coordinate encoding**: the normalized centroid coordinates $(x, y)$ (i.e., $x, y \in [-1, 1]$)) for each patch are encoded using Fourier features. For $N_f$ frequency bands (which in our case were set to $N_f = 64$ and a maximum frequency of 224), the features for each coordinate are $[\sin(s_0 \pi x), \cos(s_0 \pi x), \ldots, \sin(s_{N_f-1} \pi x), \cos(s_{N_f-1} \pi x)]$, where $s_k$ are linearly spaced frequencies. These are concatenated with the original coordinates and projected to the model's embedding dimension $D_e$.

- **Variable-specific embeddings**: distinct embedding layers are used for different categories of variables (e.g., surface, atmospheric, species) to distinguish their semantic meanings. These include dedicated embeddings for different atmospheric pressure levels and individual species channels.

- **Temporal embeddings**: both the absolute timestamp of the input and the forecast lead time $\delta t$ are encoded using sinusoidal functions and projected through separate linear layers.

The initial patch tokens are then combined by summing them with these various embeddings. The resulting feature-rich tokens $\mathbf{T}_t \in \mathbb{R}^{N_{total} \times D_e}$ (where $N_{total}$ is the total number of tokens generated across all patches and variable types/levels), form theinput to the Perceiver IO's attention mechanisms.

### A.3.2 PERCEIVER IO LATENT TRANSFORMATION

The encoder maps the input tokens $\mathbf{T}_t$ to a fixed-size latent array $\mathbf{Z}_t \in \mathbb{R}^{N_l \times D_e}$ using a two-stage process. First, a set of $N_l$ learnable latent query vectors distill information from the input tokens via cross-attention:

$$\text{CrossAttn}(\mathbf{Q}_{lat}, \mathbf{K}_T, \mathbf{V}_T) = \text{softmax}\left(\frac{\mathbf{Q}_{lat}\mathbf{K}_T^\top}{\sqrt{d_k}}\right)\mathbf{V}_T \tag{11}$$

where $\mathbf{Q}_{lat}$ are derived from the learnable queries, and $\mathbf{K}_T, \mathbf{V}_T$ are the keys and values derived from the input tokens $\mathbf{T}_t$, and $d_k$ is the dimension of the keys. The resulting latent array is then processed through a stack of self-attention layers (a Transformer tower) to allow latent tokens to interact and refine the representation. To manage computational load, this module employs Grouped-Query Attention (GQA) and standard regularization techniques like Layer Normalization and Dropout.

## A.4 THE BIOANALYST BACKBONE

The backbone ($\mathcal{B}$) serves as the neural simulation engine, taking the latent representation $\mathbf{Z}_t$ from the encoder and predicting the state for the next time step, $\mathbf{Z}'_{t+1}$. It was designed as 3D Swin Transformer architecture, structured as a U-Net. This design includes an encoder path that progressively downsamples the latent representation and a decoder path that symmetrically upsamples it, with skip connections linking corresponding stages to preserve high-resolution details.

### A.4.1 BACKBONE ENCODER PATH

The Swin Transformer backbone takes in $\mathbf{Z}_t$, a latent tensor including temporal information. $\mathbf{Z}_t$ consists of $N_l$ tokens arranged in a 3D latent grid ($N_{ld} \times N_{lh} \times N_{lw}$), with each token holding $D_e$ features.

The tensor passes through multiple encoder stages, each made of Swin Transformer blocks that apply self-attention across latent features. After each stage (except the deepest one – the "bottleneck"), patch merging halves spatial dimensions (depth, height, width) and doubles feature dimensionality, enabling coarser feature extraction.

The output features from each encoder stage, specifically the features before the path merging operation, are preserved. The preserved features are then passed to the corresponding stages in the decoder path via skip connections.

### A.4.2 BACKBONE DECODER PATH

The decoder starts from the bottleneck features and mirrors the encoder structure. Each decoder stage (except the first) begins with patch splitting, which doubles spatial resolution and halves feature dimensionality, preapring features for fusion with encoder outputs.

Skip connections combine upsampled decoder features with matching encoder outputs via element-wise addition, except at the highest resolution, where concatenation is used. This concatenated output is linearly projected to restore the feature dimension $D_e$.

Each stage then applies Swin Transformer blocks. The final decoder output, $\mathbf{Z}'_{t+1}$, matches the input $\mathbf{Z}_t$ in shape ($N_l \times D_e$) and represents the predicted next latent state.

### A.4.3 HIERARCHICAL PROCESSING WITH SHIFTED WINDOWS

As it could be noticed, the core computational unit within both the encoder and deocoder paths of the U-Net backbone is the Swin Transformer block. The Swin Transformer processes latent volumes through a series of these blocks. The number of blocks per stage and the number of the attention heads per block are configurable hyperparameters.

The following characteristics of the Swin Transformer blocks can be considered:

- **Windowed Multi-Head Self-Attention (W-MSA)**: self-attention is computed within local 3D windows (e.g., of size $W_D \times W_H \times W_W$, where $W_D, W_H, W_W$ are window dimensions for latent depth, latent height, and latent width respectively). This reduces computation compared to global self-attention, as attention is restricted to non-overlapping local windows.

- **Shifted Window Multi-Head Self-Attention (SW-MSA)**: to allow for cross-window connections, consecutive Swin Transformer blocks, alternate between regular W-MSA and SW-MSA. In SW-MSA, the window configuration is shifted by half a window size relative to the previous layer. This cyclic shift ensures that the creation of a larger receptive field over layers.

- **Relative position bias**: relative position biases are added to the attention scores, potentially improving generalization across different locations within the windows.

- **Multi Layer Perceptron (MLP) layers:** each attention module is followed by a 2-layer MLP with Gaussian Error Linear Unit (GELU).

### A.5 THE BIOANALYST DECODER

The BioAnalyst decoder ($\mathcal{D}$) translates the predicted latent state $\mathbf{Z}'_{t+1}$ from the backbone back into a high-resolution, multi-modal grid of observable variables $\hat{\mathbf{X}}_{t+1} \in \mathbb{R}^{\mathcal{H} \times \mathcal{W} \times C_{out}}$. Similar to the encoder, the decoder uses a Perceiver IO core, allowing for flexible and targeted generation of outputs.

### A.5.1 OUTPUT QUERY FORMULATION

The decoder uses a set of specific, learnable output queries, each targeting a specific variable at a given spatial location and time. For each target variable map (e.g., surface temperature or species distribution), a query vector $\mathbf{q} \in \mathbb{R}^{D_e}$ is formed by combining:

- **Target variable embedding**: identifies the variable to predict (e.g., 'surface temp', 'species X').

- **Spatial coordinate embedding**: spatial positions over the $(\mathcal{H}, \mathcal{W})$ grid are encoded via Fourier features, projected to $D_e$, and interpolated to match the number of output queries ($N_q$), yielding a shared spatial tensor of shape $N_q \times D_e$.

- **Temporal embedding**: encodes the forecast time step $t + 1$ using a lead time embedding and optionally, an absolute time encoding.

- **Atmospheric level/Species index embedding** (if applicable): distinguishes pressure levels or species indices for level-specific or species-specific outputs.

These components are summed to form each query vector. Collectively, the queries form a query array $\mathbf{Q} \in \mathbb{R}^{N_q \times D_e}$ (where $N_q$ is the total number of distinct output variable maps to be generated), covering all defined outputs (surface, single-level, atmospheric, species, land etc.).

After attending to $\mathbf{Z}'_{t+1}$, the decoder produces output embeddings of shape $N_q \times D_e$. These are projected through task-specific layers into flat variable maps, then reshaped into $\mathcal{H} \times \mathcal{W}$ grids—one for each target variable, level, or species.

### A.5.2 CROSS-ATTENTION WITH BACKBONE OUTPUT

The output queries $\mathbf{Q}$ attend to the final latent state $\mathbf{Z}'_{t+1}$ produced by the Swin Transformer backbone. This cross-attention mechanism allows each query to selectively extract the relevant information from the dense latent representation needed to predict its specific target. The attention operation is analogous to that in the encoder:

$$\hat{\mathbf{Y}}_{t+1} = \text{CrossAttn}(\mathbf{Q}, \mathbf{K}_{Z'}, \mathbf{V}_{Z'}) = \text{softmax}\left(\frac{\mathbf{Q}\mathbf{K}_{Z'}^T}{\sqrt{d_k}}\right)\mathbf{V}_{Z'} \tag{12}$$

where $\mathbf{K}_{Z'}$ and $\mathbf{V}_{Z'}$ are keys and values derived from the backbone's output $\mathbf{Z}'_{t+1}$. The result, $\hat{\mathbf{Y}}_{t+1} \in \mathbb{R}^{N_q \times D_e}$, is an array where each row corresponds to an output query and contains the decoded information in the embeddings dimension.

Essentially, the Perceiver IO architecture used for the decoder focuses on the cross-attention between the output queries and the backbone's output latent state.

### A.5.3 PROJECTION AND RESHAPING TO FINAL OUTPUT

The Perceiver IO decoder outputs a sequence of embeddings, $\hat{\mathbf{Y}}_{t+1}$, where each embedding must be projected to the actual value of its target variable. A variable-specific linear projection maps each $D_e$-dimensional embedding to the appropriate output shape. For scalar outputs (e.g., temperature at a location), this means projecting to a single value.

Dedicated projection layers handle each variable group, converting embeddings into structured outputs. For example, atmospheric variables are projected into tensors shaped by batch size, number of variables, number of pressure levels, height, and width. Species-related outputs include an extra species dimension.

After projection, all outputs are reshaped and organized into the final multi-modal grid $\hat{\mathbf{X}}_{t+1} \in \mathbb{R}^{\mathcal{H} \times \mathcal{W} \times C_{out}}$, where $C_{out}$ is the total number of predicted variable channels.

### A.6 POSITIONAL, TEMPORAL, AND VARIABLE-SPECIFIC ENCODINGS

Effective representation of spatial, temporal, and categorical information is highly-important for BioAnalyst to interpret inputs and generate accurate, context-aware predictions. This sub-appendix provides more details into the specific encoding schemes that were used.

### A.6.1 TEMPORAL ENCODING SCHEMES

BioAnalyst encodes two temporal aspects: the absolute time of an observation and the forecast lead time.

- **Absolute time encoding**: The calendar date and time of an input (or decoder target) time step $t$ are converted to a scalar value $\tau$, then encoded using sinusoidal functions. For embedding dimension $D_e$, the resulting vector $\mathbf{e}_{time} \in \mathbb{R}^{D_e}$ has components: $(\mathbf{e}_{time})_{2i} = \sin(\tau/10000^{2i/D_e})$ and $(\mathbf{e}_{time})_{2i+1} = \cos(\tau/10000^{2i/D_e})$ for $i \in [0, D_e/2 - 1]$. This encoded vector is then processed by a learned linear layer to match $D_e$ and allow for learnable adaptation.

- **Lead time encoding:** the forecast lead time, $\Delta t = t_{forecast} - t_{input}$, is also encoded.This scalar value (e.g., 2 months) undergoes the same sinusoidal encoding as above and is then projected using a separate learned linear layer. This informs the model of how far into the future it is forecasting.

Both the encoder and decoder components use these time encodings, adding them to the patch or query embeddings to provide temporal context.

### A.6.2 Variable-Specific and Categorical Feature Embeddings

To differentiate between the various input modalities and their specific characteristics, BioAnalyst uses learned embeddings for different categories of data:

- **Variable type embeddings**: each input variable (e.g., 2m temperature, species extinction risk) gets a unique, learnable embedding of size $D_e$. Separate linear layers are used for different variable groups, projecting the patchified data (which implicitly includes variable identity due to how data is batched and fed) into the shared embedding space.
- **Atmospheric level embeddings**: for atmospheric variables variables across pressure levels (e.g., 50 hPa, 500 hPa), each level's spatial data is tokenized and projected using a shared linear layer. These level-specific tokens are then concatenated with others, letting the model differentiate vertical context via token position.
- **Species index embeddings**: similarly to atmospheric levels, for multi-species variables, each species channel is tokenized and projected via a shared layer. These species-specific tokens are concatenated to enable attention layers to learn species-aware features.

All embeddings are learned during training and added to the patch tokens before being fed into Perceiver IO, providing spatial, temporal, and semantic context for forecasting.

### A.7 Data Normalization

BioAnalyst normalizes all variables before processing them in the encoder and unnormalizes the outputs of the decoder to produce the final predictions. All the variables are normalized separately, and the variables which have more levels are normalized per-level (e.g., species distributions and atmospheric variables). For the normalization and denormalization, we compute statistics across the whole dataset by collecting mean values and standard deviations. The relationship between normalized and unnormalized variables is the following:

$$\mathbf{X}_{v,i,j,\text{normalized}}^t = \frac{\mathbf{X}_{v,i,j}^t - \text{centre}_v}{\text{scale}_v} \tag{13}$$

### A.8 Emulation stage

Given a system state $\mathbf{X}_t$ at time $t$, BioAnalyst goal is to predict the next state $\mathbf{X}_{t'}$ at time $t' > t$. In the common single-step forecast scenario, following (Bodnar et al., 2024), we define a simulator function

$$\Phi : (\mathbf{X}_{t-1}, \mathbf{X}_t) \to \hat{\mathbf{X}}_{t+1}, \tag{14}$$

which given two consecutive system states $\mathbf{X}_{t-1}$ and $\mathbf{X}_t$, predicts the future state $\hat{\mathbf{X}}_{t+1}$. Once we learn $\Phi$, we can roll out predictions over extended horizons in an autoregressive manner. Concretely, after setting $\hat{\mathbf{X}}_t = \mathbf{X}_t$ and $\hat{\mathbf{X}}_{t-1} = \mathbf{X}_{t-1}$, we can write:

$$\hat{\mathbf{X}}_{t+k} = \Phi(\hat{\mathbf{X}}_{t+k+2}, \hat{\mathbf{X}}_{t+k-1}), \text{for } k = 1, 2, ... \tag{15}$$

so, the next predicted state depends on the two most recent states, specifically the last real or predicted step. This repeated application of $\Phi$ is referred to as an iterative or autoregressive rollout.

## B  DATASET

This section provides a detailed description of the data used to pre-train and finetune BioAnalyst.

### B.1  INTRODUCTION

BioAnalyst is pre-trained on a part of BioCube as discussed at section 3.2, using a Batch structure throughout with statistics available at Table 5. Our decision to use $0.25°$ grid resolution is motivated by a substantial body of work in macroecology and conservation planning that operates at comparable or even coarser resolutions when addressing broad-scale questions (e.g. global biodiversity risk, identification of priority regions, assessment of conservation networks) (Engemann et al., 2015). For example, recent global analyses of climate-change risks to terrestrial biodiversity and macroecological diversity patterns have used 25–50 km grid cells, arguing that such coarse units are appropriate for capturing broad climatic and biogeographic gradients relevant to policy (Roll et al., 2017). Similarly, assessments of Key Biodiversity Areas and other large-scale conservation prioritization exercises often use 25–50 km planning units. Thus, a $0.25°$ grid is well aligned with the spatial grain at which many national or continental assessments are currently performed (Farooq et al., 2023).

### B.2  VARIABLE GROUPS

For our data mixture we have used the 11 variable groups $V$ with their corresponding variables $v$ available on Table 2.

This variables selection is inspired and extracted from various works around species distribution modelling, habitat assesement and ecosystem modelling with classical and Machine Learning methods.

More specific, climatic energy and moisture variables like temperature and humidity reflect thermal nichies where humidity gives vapour-pressure deficit a key plant and anthropogenic stressor (Zuquim et al., 2020). Precipitation indicates the water balance which is the strongest global predictor after temperature (Gutiérrez-Hernández & García, 2021). Radiation on various lengths (short and long) drives photosynthesis and evapotranspiration and has showed improvement in Net Primary Productivity (NPP)-linked biodiversity models (Brown et al., 2020a). Snow variables at high latitudes and mountains control growing season length and habitat suitability.

For edaphic water and temperature status, we selected to look into soil moisture which is a direct proxy for plant water stress and root-zone dynamics and has a strong interactive effect with species richness (Xu et al., 2022). Soil temperature provides germination cues linked with soil microbial activity while variables like potential evapotranspiration and surface latent heat flux are need for water deficit estimation (Schönauer et al., 2024).

For vegetation, we select Normalized Difference Vegetation Index (NDVI) that quantifies the health and density of vegetation and is a strong indicator of the greenness of the biomass. Additionally, we select a series of indicators that quantify the monthly changes of land, agriculture (arable and cropland).

Finally, for species we select 2 variables to work with. The first is the Red List Index (RLI) which is an indicator of the changing state of global biodiversity and defines the conservation status of major species group and measures trends in extinction risk over time. The second is the species occurrences records for 6 major categories that is highly ranked in importance from European Union. More specific large carnivores, farmland birds, wild pollinators, herpetofauna, invasive alien species (IAS) and Mediterranean endemics together span the core biodiversity priorities currently driving European nature policy and funding. Large carnivore management has become a political flashpoint: the Nature Restoration Regulation explicitly references "conflict species" such as the wolf, and the European Parliament voted in May 2025 to downgrade the wolf from strict to general protection under the Habitats Directive (Commission, 2022). Farmland birds remain the EU's headline biodiversity indicator; the European Environment Agency reports a $40\%$ decline in the Farmland Bird Index since 1990, underscoring the need for landscape-scale restoration (Agency, 2024). Wild pollinators stand at the centre of the revised EU Pollinators Initiative ("A new deal for pollinators"), which commits all Member States to continent-wide monitoring and trend reversal by 2030

Table 2: Overview of variable groups used as input features for the pre-training and fine-tuning of BioAnalyst.

| Variable group | Variables | Dimensionality |
|---|---|---|
| Surface variables | `t2m`, `msl`, `slt`, `z`, `u10`, `v10`, `lsm` | [160, 280, 7] |
| Edaphic variables | `swvl1`, `swvl2`, `stl1`, `stl2` | [160, 280, 4] |
| Pressure levels (atmospheric) | 1000, 925, 850, 700, 600, 500, 400, 300, 250, 200, 150, 100, 50 | 13 |
| Atmospheric variables | `z`, `t`, `u`, `v`, `q` | [160, 280, 5, 13] |
| Climate variables | `smlt`, `tp`, `csfr`, `avg_sdswrf`, `avg_snswrf`, `avg_snlwrf`, `avg_tprate`, `avg_sdswrfcs`, `sd`, `t2m`, `d2m` | [160, 280, 11] |
| Miscellaneous variables | `avg_slhtf`, `avg_pevr` | [160, 280, 2] |
| Vegetation variables | NDVI | [160, 280, 1] |
| Land variables | Land (percentage of total land area) | [160, 280, 1] |
| Agriculture variables | Agriculture, Arable, Cropland (percentage of land area) | [160, 280, 3] |
| Redlist variables | RLI (Red List Index for biodiversity state) | [160, 280, 1] |
| Forest variables | Forest (percentage of land area) | [160, 280, 1] |
| Species variables | Species occurrence records | [160, 280, 28] |

(European Commission, Directorate-General for Environment, 2023). IAS require surveillance and early-warning systems under Regulation (EU) 1143/2014, which obliges Member States to establish national monitoring and rapid-response mechanisms (European Commission, 2014). Finally, herpetofauna and Mediterranean endemics receive targeted LIFE funding and are focal taxa in Article 17 conservation-status reporting under the Habitats Directive, anchoring them in the EU's mandatory assessment cycle. Table 3 lists in detail all the species from the above categories that are used for training BioAnalyst.

### B.3 BUILD BATCHES

We have developed a pipeline of systematic data batch construction to enable learning across diverse and temporally aligned environmental and biodiversity signals. Each batch combines multiple data modalities over a fixed temporal window of one calendar month, including both the beginning and end time points. Its design ensures compatibility with geospatial deep learning models that require structured tensors across space, time, and modality. The pipeline creates batches with a temporal resolution of two consecutive monthly slices (e.g., January and February 2001). Spatially, it follows a fixed grid at $0.25° \times 0.25°$ resolution, matching the Copernicus ERA5 grid and covering the full anticipated range of longitudes and latitudes within Europe. All variables are re-projected via appropriate transformations to this common spatial resolution to ensure alignment across modalities. The pipeline supports the data sources and modality groups introduced in Section B.2.

Table 3: Approximate total occurrences (2000–2020) and GBIF species ID to scientific name mappings for six major species categories

| Major category | Species ID | Scientific name (common) | Total occurrences |
|---|---|---|---|
| Farmland birds | 8077224 | *Alauda arvensis* (skylark) | 2.5 M |
| | 2491534 | *Emberiza citrinella* (yellowhammer) | 2.5 M |
| | 2473958 | *Perdix perdix* (grey partridge) | 400 k |
| | 4408498 | *Crex crex* (corncrake) | 140 k |
| | 9809229 | *Sturnus vulgaris* (common starling) | 5.0 M |
| Herptiles | 2431885 | *Triturus cristatus* (great crested newt) | 149 k |
| | 8909809 | *Emys orbicularis* (European pond turtle) | 39 k |
| | 2430567 | *Pelobates fuscus* (spadefoot toad) | 12 k |
| Invasive & alien | 8002952 | *Ambrosia artemisiifolia* (common ragweed) | 87 k |
| | 2437394 | *Callosciurus erythraeus* (Pallas's squirrel) | 900 k |
| | 3034825 | *Heracleum mantegazzianum* (giant hogweed) | 181 k |
| | 2891770 | *Impatiens glandulifera* (Himalayan balsam) | 422 k |
| | 5218786 | *Procyon lotor* (raccoon) | 36 k |
| Large carnivores | 5219173 | *Canis lupus* (grey wolf) | 22.5 k |
| | 2433433 | *Ursus arctos* (brown bear) | 5.8 k |
| | 2435240 | *Lynx lynx* (Eurasian lynx) | 34.9 k |
| | 5219219 | *Canis aureus* (golden jackal) | 1.4 k |
| | 5219073 | *Gulo gulo* (wolverine) | 5.7 k |
| Mediterranean | 2435261 | *Lynx pardinus* (Iberian lynx) | 436 k |
| | 5844449 | *Aquila fasciata* (Bonelli's eagle) | 55 k |
| | 2441454 | *Testudo hermanni* (Hermann's tortoise) | 34 k |
| | 2434779 | *Monachus monachus* (Med. monk seal) | 137 k |
| | 8894817 | *Caretta caretta* (loggerhead sea turtle) | 6.5 k |
| Pollinators | 1340503 | *Bombus terrestris* (buff-tailed bumblebee) | 266 k |
| | 1340361 | *Bombus hyperboreus* (Arctic bumblebee) | 325 k |
| | 1898286 | *Vanessa atalanta* (red admiral) | 2.0 M |
| | 1920506 | *Pieris brassicae* (large white) | 1.8 M |
| | 1536449 | *Episyrphus balteatus* (marmalade hoverfly) | 0.01 k |

The batching process is fully deterministic: for a given input dataset and time window, it produces outputs without random variability. This design enables consistent benchmarking across runs and supports long-term model evaluation on fixed test splits. Furthermore, the use of non-overlapping monthly intervals ensures temporal independence between batches, which is essential for forecasting and change detection tasks.

Every input dataset is preprocessed and harmonised based on the following principles:

- Longitude coordinates are wrapped into the interval $(-180°, 180°]$ to ensure consistency across global datasets.

- Timestamps are standardised to `datetime64` objects with monthly resolution.

- Latitude and longitude coordinates are snapped to the $0.25°$ grid to match the spatial resolution of the batch format.

- Missing values are imputed with zeros, enabling compatibility with models that do not natively support NaN values.

Table 4: ERA5 variable names, definitions and units (Copernicus Climate Change Service (C3S), 2017; Hersbach et al., 2020)

| Variable | Units | Description |
| --- | --- | --- |
| swvl1 | $\mathrm{m^3\,m^{-3}}$ | Volumetric soil water content, layer 1 (0 cm depth) |
| swvl2 | $\mathrm{m^3\,m^{-3}}$ | Volumetric soil water content, layer 2 (7 cm depth) |
| stl1 | K | Soil temperature, layer 1 (0 cm depth) |
| stl2 | K | Soil temperature, layer 2 (7 cm depth) |
| smlt | m (water eq.) | Snow melt accumulated at surface |
| tp | m | Total precipitation (liquid + frozen) |
| csfr | $\mathrm{kg\,m^{-2}\,s^{-1}}$ | Convective snowfall rate |
| avg_sdswrf | $\mathrm{W\,m^{-2}}$ | Mean surface downwelling shortwave radiation flux |
| avg_snswrf | $\mathrm{W\,m^{-2}}$ | Mean surface net shortwave radiation flux |
| avg_snlwrf | $\mathrm{W\,m^{-2}}$ | Mean surface net longwave radiation flux |
| avg_tprate | $\mathrm{m\,s^{-1}}$ | Mean total precipitation rate |
| avg_sdswrfcs | $\mathrm{W\,m^{-2}}$ | Mean surface downwelling shortwave flux (clear sky) |
| sd | m | Snow depth |
| t2m | K | 2 m air temperature |
| d2m | K | 2 m dew point temperature |
| msl | Pa | Mean sea level pressure |
| slt | code | Soil type classification code |
| z | $\mathrm{m^2\,s^{-2}}$ | Geopotential |
| t | K | Air temperature (pressure levels) |
| u | $\mathrm{m\,s^{-1}}$ | Eastward wind component (pressure levels) |
| v | $\mathrm{m\,s^{-1}}$ | Northward wind component (pressure levels) |
| u10 | $\mathrm{m\,s^{-1}}$ | 10 m eastward wind component |
| v10 | $\mathrm{m\,s^{-1}}$ | 10 m northward wind component |
| q | $\mathrm{kg\,kg^{-1}}$ | Specific humidity (pressure levels) |
| lsm | 0/1 | Land–sea mask (0 = sea, 1 = land) |
| avg_slhtf | $\mathrm{W\,m^{-2}}$ | Mean surface latent heat flux |
| avg_pevr | $\mathrm{kg\,m^{-2}\,s^{-1}}$ | Mean potential evaporation rate |

To maintain modularity and extensibility, the pipeline separates data loading logic per modality (e.g., `_load_era5`, `_load_csv`, `_load_species`), with all outputs standardised into spatially and temporally aligned `xarray.Dataset` objects. This design supports the seamless addition of future data types such as Sentinel-2 imagery, elevation, or anthropogenic indicators without altering the core batch assembly logic.

The batching pipeline is optimised to handle large-scale datasets efficiently. ERA5 NetCDF files are processed using `xarray` with chunking enabled, and CSV files are filtered and indexed spatially using precomputed 0.25° grid maps. This architecture allows scaling to continental datasets and long temporal ranges without excessive memory usage.

`Xarray` reads ERA5 NetCDF files, combines them by coordinates, and retains exactly two calendar months per batch. If multiple temporal slices exist within the same month, only the earliest one is kept to reduce redundancy. Land, agriculture, or vegetation CSV files are ingested and parsed into month-specific rasters. The pipeline supports both common CSV structures:

- Layout A: includes a `Variable` column and individual year columns (e.g., `2000`, `2005`).

- Layout B: includes variable-year columns directly (e.g., `NDVI_2020`, `Land_2015`).

If expected variables or time slices are missing in a particular file, the system logs the missing entries and fills them with zeros. This ensures that tensor dimensions remain consistent across batches, even when data coverage is sparse for certain modalities or regions.

Species data are extracted from Parquet files. Each species is treated as an independent raster tensor per month, based on the reported `Distribution` value. A master species list is inferred from

Table 5: Statistics of the constructed data batches used in the pre-training of BioaAnalyst

| Attribute | Value |
|---|---|
| Grid sampling resolution | 0.25° |
| Global grid size | 1440 (lon) × 720 (lat) = 1.036.800 cells |
| Europe grid size | 280 (lon) × 160 (lat) = 44.800 cells |
| Latitude bounds | [32°, 72°] |
| Longitude bounds | [−25°, 45°] |
| Time range | 01–01–2000 to 01–06–2020 |
| Number of batches | 233 |
| Batch size (average) | ∼43 MB |
| Total storage volume | 10 GB |
| Temporal resolution per batch | 2 months |
| Pressure levels included | 13 |
| Species per batch | 28 |
| Total data points | 5.062.400 |

the available data Table 3 to ensure consistent dimensionality across batches. This design enables both single-species and multi-species modelling and supports spatially explicit predictions of species presence.

All variables are converted into PyTorch tensors. Scalar geospatial variables (e.g., temperature, NDVI) are stored in tensors of shape $(2, H, W)$, where $H$ and $W$ are the grid height and width and 2 accounts for the consecutive timestamps. Pressure-level variables (e.g., geopotential, wind at altitude) are stored as $(2, C, H, W)$ where $C$ is the number of pressure levels. Species presence tensors are organised as one tensor per species with shape $(2, H, W)$. In cases where pressure-level variables are included, their corresponding pressure levels are saved in the batch metadata.

Each batch includes a metadata dictionary capturing the timestamp window, grid specification (latitude and longitude), list of included species, and pressure levels (if applicable). The final batch is a structured dictionary with modality keys representing the various variable categories (e.g., surface_variables, agriculture_variables, species_variables), each containing variable-specific tensors. The complete batch is serialised and saved to disk in PyTorch's binary .pt format, enabling efficient loading during model training without repeating and preprocessing operation.

### B.4 FINE-TUNE DATASETS

For task-specific finetuning we have used two different datasets for the corresponding tasks respectively. More specific we have used:

**GeoLifeCLEF24:** GeoLifeCLEF is benchmark dataset with a training set of close to 5 million plant occurrences in Europe (single-label, presence-only data) as well as a validation set of about 5000 plots and a test set with 20000 plots, with all the present species (multi-label, presence-absence data) (Joly et al., 2024) from 2017 to 2021. A visual depiction of how the data look like can be found on Figure 8.

**CHELSA v2.1:** CHELSA (Climatologies at High resolution for the Earth's Land Surface Areas) delivers global grids of mean, minimum and maximum 2 m air temperature (tas, tasmin, tasmax) and precipitation rate (pr) at 30-arc-second resolution ($\sim 1$ km) from 1979 - present (we use 2000-2019 monthly records). Temperature fields are produced through lapse-rate–based statistical downscaling of ERA-5/ERA-Interim reanalyses, while precipitation is downscaled with an orographic algorithm that incorporates wind fields, boundary-layer height and subsequent bias-correction with Global Precipitation Climatology Centre (GPCC) gauge data, yielding markedly improved representation of mountain rain-shadows and thermal gradients compared with coarser products (Karger et al., 2017). These high-fidelity temperature and precipitation layers form the foundation for CHELSA's 19 bioclimatic derivatives and for time-series forcings such as CHELSA-W5E5, and are now a de-facto standard in species-distribution modelling, trait–environment analyses, hydrological and vegetation-

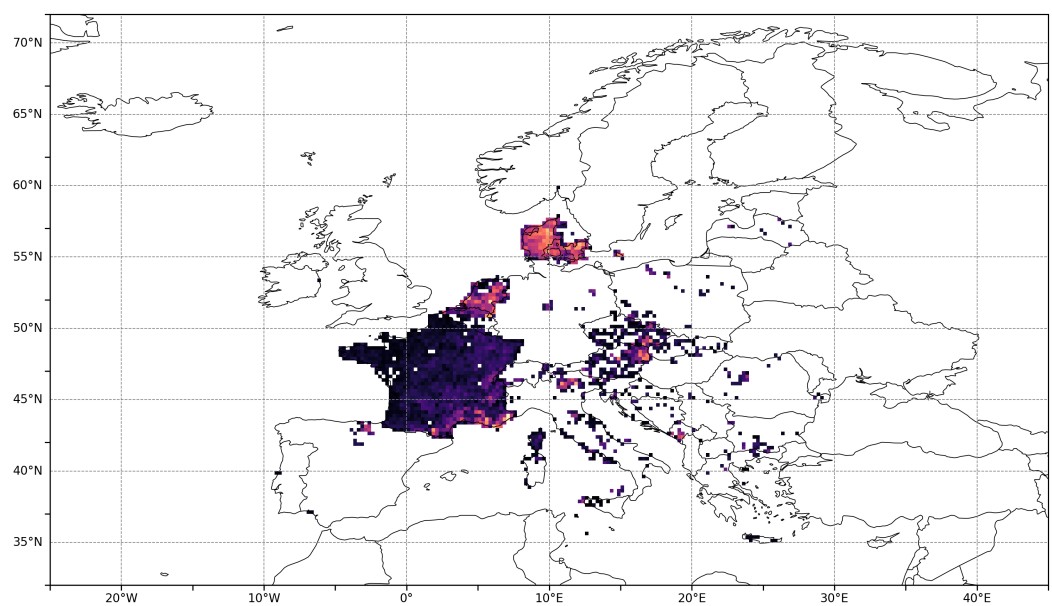

Figure 8: Yearly aggregated plant species richness per $0.25° \times 0.25°$ grid across Europe ($35 - 60°$ N, $10°$ W $- 30°$ E) for the GeoLifeCLEF24 survey data from 2018-2021 with 5 million occurrences. Lighter colors show higher abundance.

dynamics models, and climate-change impact assessments. Because ecological responses to climate are often threshold-driven and spatially heterogeneous, the $\sim 1$ km detail of CHELSA allows modellers to resolve local refugia, elevational turnover and fine-scale moisture stress that coarser climatologies fail to capture, thereby increasing predictive accuracy and reducing uncertainty across taxa and regions (Karger et al., 2020).

Table 6: Model card

| model | patch_size | num_heads | embed_dim | depth | swin_size | model size |
|-------|-----------|-----------|-----------|-------|-----------|-----------|
| Medium | 4 | 12 | 384 | 6 | medium | 440M |
| Large | 8 | 16 | 512 | 10 | large | 700M |

Table 7: Swin-backbone configurations for large and medium model sizes

| Parameter | Large | Medium |
|-----------|-------|--------|
| encoder_depths | [2, 2, 2] | [2, 2] |
| encoder_num_heads | [8, 16, 32] | [8, 16] |
| decoder_depths | [2, 2, 2] | [2, 2] |
| decoder_num_heads | [32, 16, 8] | [16, 8] |
| window_size | [1, 4, 5] | [1, 1, 1] |
| mlp_ratio | 4.0 | 4.0 |
| qkv_bias | True | True |
| drop_rate | 0.1 | 0.0 |
| attn_drop_rate | 0.1 | 0.0 |
| drop_path_rate | 0.1 | 0.1 |

## C  IMPLEMENTATION DETAILS

This appendix provides detailed information about the model card, the hyperparameters the training recipe, the software used and the methods used for efficient scaling, as a supplement to sections 3.3 and 3.4

### C.1  MODEL CARD

We have implemented and star training 2 versions of BioAnalyst, one Small sized with 440M parameters and one Medium with 980M parameters. The complete model configuration can be found on Table 6 and Table 7. The idea behind these configurations is the increasing and decreasing sizes of the kernel dimensions, following our U-Net style backbone architecture.

Furthermore, to enhance performance and computational efficiency, particularly given the high dimensionality of the input and output data, several techniques are used. Group-Query Attention (Ainslie et al., 2023) is used within the Perceiver IO attention layers to reduce the memory footprint associated with key-value projections during cross-attention. Additionally, standard regularisation techniques such as Layer Normalisation and Dropout are applied throughout the Perceiver modules. The Swin Transformer backbone makes use of stochastic depth (Huang et al., 2016) to improve generalisation by randomly skipping the residual branch during training, thus reducing the depth on a per-sample basis.

### C.2  SOFTWARE

For the development of BioAnalyst we used the Python programming language (Foundation, 2025) and developed the required modules on PyTorch's neural network library (Paszke et al., 2019) and PyTorch Lightning (Falcon & The PyTorch Lightning team, 2019). For visualisation we employed the Streamlit package (Team, 2025). For data operations we used xarray package (Hoyer & Hamman, 2017).

### C.3  TRAINING AND SCALING

BioAnalyst has been trained for 1000 epochs - 80.000 gradient steps with a batch size of 1, using the AdamW optimiser (Loshchilov & Hutter, 2019) with cosine-annealing learning rate schedule with periodic warm restarts (Loshchilov & Hutter, 2017). We used a starting learning rate of 0.00005 and

weight decay of $0.000005$ with $T_{periodic} = 8000$ gradient steps. We train the complete architecture together, both encoder-decoder and backbone.

**Variable weighting**

To balance the loss during pre-training and subsequent finetuning tasks we assign individual weights to each variable for every variable group. Our weighting scheme is inspired by the works of (Bodnar et al., 2024; Piedallu et al., 2013; Zomer et al., 2022; Carbognani et al., 2012; Coelho et al., 2023; Maharjan et al., 2022; Bates et al., 2025) and are reported on Table 8.

**Training Loss**

As discussed in section 3.3 our pre-training objective is the temporal difference error $\mathcal{L}_{TD}$ which is

$$\mathcal{L}_{TD} = ||\hat{\Delta x_t^v} - (x_{t+1}^v - x_t^v)|| \tag{16}$$

In addition, during training we weight each variable's loss with the weights we defined before at Table 8

$$\mathcal{L}_{\text{TD}}(t) \; = \; \sum_{v \in \mathcal{V}} w_v \left\| \hat{\Delta x_t^v} - \left( x_{t+1}^v - x_t^v \right) \right\|_1, \tag{17}$$

### C.4 FINETUNING TASKS

#### C.4.1 ROLL-OUT FINETUNING

BioAnalyst has been pre-trained to predict $x_{t+1}$ by providing as input $[x_{t-1}, x_t]$. During fine-tuning and more specific, roll-out fine-tuning, BioAnalyst was extended to predict 6 and 12 steps ahead in 2 stages. At the first stage, we fine-tune the complete BioAnalyst, adding VeRA adapters on the backbone for 12000 steps with a horizon of 6 time-steps (6 months), using constant learning rate of $0.00003$, weight decay of $0.000003$ and store the weights. In the second stage we load the previously stored weights and continue the roll-out fine-tuning for 12000 time-steps more, with a horizon of 12 steps (1 year), using constant learning rate of $0.00001$ and weight decay of $0.000001$. Fine-tuning with a mix of variable horizons is a technique used on Aurora (Bodnar et al., 2024), PrithviWxC (Schmude et al., 2024) and Aardvak (Allen et al., 2025), offering improved long-horizon forecasting capabilities, which we also observe ourselves and are highlighted on Figure 3.

#### C.4.2 ABIOTIC LINEAR PROBING: RECOVERING SEASONAL CLIMATE STRUCTURE

To probe the fidelity of BioAnalyst's pretrained environmental embeddings, we train a regression head to predict monthly climate values from the CHELSA v2.1 dataset against BioAnalyst's decoder outputs. Specifically, we predict the monthly mean temperature (**tas**) and total precipitation (**pr**) over Europe ($0.25°$ grids) for the years 2000 to 2019 (BioAnalyst is trained on data up to 2018). We reconstruct the decoder outputs for the same variables from BioAnalyst. For comparison, we used Aurora's decoder predictions for 2-meter temperature. This creates a linear probing setup for evaluating how well the pretrained BioAnalyst latent representations encode climatically relevant information. By comparing the decoded reconstructions to the downsampled CHELSA targets, this linear probing setup assesses the alignment between learned representations and real-world climate signals, without updating the pretrained weights.

For each grid cell in the study area, the model predicts a 24-dimensional target vector representing the full annual cycle of temperature and precipitation (see Figure 7). We use mean squared error as the training objective and evaluate performance using RMSE, $R^2$, and monthly correlation metrics across diverse European bioclimatic zones. This probing task tests whether BioAnalyst's pretrained latent space captures fine-grained abiotic structure, particularly seasonal variation, that is critical for ecological processes. As the encoder remains frozen, this setup isolates the representational quality of the pretrained embeddings, without the confounding effects of adaptation or fine-tuning.

Table 8: Variable-specific weights $(w_v)$ used in the loss function

| Variable group $(V)$ | Variable $(v)$ | Weight $(w_v)$ |
|---|---|---|
| Surface | t2m | 2.50 |
| | msl | 1.50 |
| | slt | 0.80 |
| | z | 1.00 |
| | u10 | 0.77 |
| | v10 | 0.66 |
| | lsm | 1.20 |
| Edaphic | swvl1 | 1.10 |
| | swvl2 | 0.90 |
| | stl1 | 0.70 |
| | stl2 | 0.60 |
| Atmospheric (p-levels) | z_pl | 2.80 |
| | t_pl | 1.70 |
| | u_pl | 0.87 |
| | v_pl | 0.60 |
| | q_pl | 0.78 |
| Climate | smlt | 1.00 |
| | tp | 2.20 |
| | csfr | 0.60 |
| | avg_sdswrf | 0.90 |
| | avg_snswrf | 0.70 |
| | avg_snlwrf | 0.50 |
| | avg_tprate | 2.00 |
| | avg_sdswrfcs | 0.50 |
| | sd | 0.90 |
| | t2m_clim | 2.50 |
| | d2m | 1.30 |
| Vegetation | NDVI | 0.80 |
| Land cover | Land | 0.60 |
| Agriculture | Agriculture | 0.40 |
| | Arable | 0.30 |
| | Cropland | 0.40 |
| Forest | Forest | 1.20 |
| Redlist | RLI | 1.30 |
| Miscellaneous | avg_slhtf | 1.20 |
| | avg_pevr | 1.00 |
| Species | species | 10.00 |

### C.4.3 BIOTIC FINE-TUNING: FORECASTING SPECIES DISTRIBUTIONS

To assess whether BioAnalyst can learn to forecast biodiversity dynamics across time, we fine-tune the model's backbone using anonymised plant species presence-only observations from the GeoLifeCLEF24 dataset. The model is trained on occurrences of 500 species (most frequent in the

GeoLifeCLEF24 survey) across Europe from 2017 to 2020 and evaluated on its ability to predict species presence in 2021, without access to future climate or land-use data. The anonymity of the species focuses the modelling exercise on the process, rather than the species identity or phylogeny.

The model takes as input a species distribution matrix at time $t$ and is trained to predict the distribution at time $t + 1$. The input and target distributions are normalised using species-specific statistics (mean and standard deviation). The model preserves the full spatial resolution of the data, working with grid-based representations rather than individual occurrence points. Training is performed end-to-end using a combination of loss functions, including the GeoLifeClef defined F1 score (Picek et al., 2024) and root mean square error (RMSE). The custom F1 score is determined in Equation 20. These metrics indicate whether the model learns both local and global distribution patterns, and also gives a comparable score for the GeoLifeCLEF benchmark.

### C.4.4 BASELINES

We include Aurora-$0.25°$ as a strong, climate-only baseline (Bodnar et al., 2024) by attaching a simple prediction head and fine-tuning it on our task. This comparison is necessarily imperfect: Bio-Analyst is pre-trained directly on monthly biodiversity states and multimodal ecological covariates, whereas Aurora is an Earth-system foundation model trained on over a million hours of geophysical data and optimised for short-lead (6-hour) forecasting of atmospheric and oceanic variables, then fine-tuned for specific operational weather, wave and air-quality tasks. As a result, our Aurora baseline should be viewed as an indicative demonstration of how a generic climate foundation model performs when naively adapted to biodiversity prediction, rather than a definitive assessment of Aurora's capabilities. Our empirical evidence is limited to a small number of downstream biodiversity tasks and one region, and we therefore refrain from drawing general conclusions about Aurora itself.

**Joint Species Distribution Modelling**  To establish strong yet interpretable baselines for joint spatio–temporal species distribution modelling (jSDM), we consider two complementary architectures. First, we implement a non-spatial latent-factor jSDM that treats each grid cell independently and models the joint community vector with a shared multilayer perceptron (MLP) - LatentMLP. This design follows the generalized linear latent-variable / jSDM tradition (Warton et al., 2015; Ovaskainen et al., 2017), in which cross-species residual covariance is captured via low-dimensional latent factors rather than separate single-species models. In practice, we encode the presence–absence vector of all species at time $t$ into a low-dimensional community embedding and decode it into species-specific logits for time $t+1$, yielding a multi-output Bernoulli model closely aligned with recent scalable jSDM implementations such as *sjSDM* (Pichler & Hartig, 2021). This baseline provides a conceptually clean joint model that captures species co-occurrence structure but ignores explicit spatial autocorrelation.

Second, we introduce a spatio–temporal ConvLSTM jSDM that operates directly on the $C \times H \times W$ species distribution grids over consecutive years. We first project the species axis into a smaller set of learned community channels using $1 \times 1$ convolutions, then apply a convolutional LSTM over the temporal dimension to propagate information across both space and time before decoding back to species-level probabilities with a sigmoid output. ConvLSTMs were originally proposed for precipitation nowcasting as powerful spatio–temporal sequence models (Shi et al., 2015) and are now widely used in environmental forecasting. Our baseline adapts this idea to multi-species distributions, in line with recent work showing that joint deep neural networks improve community-level predictions compared to species-wise models (Brun et al., 2024) and that deep-SDM frameworks such as MALPOLON scale to thousands of species and high-resolution remote-sensing inputs (Larcher et al., 2024). Finally, by training and evaluating these baselines on GeoLifeCLEF2024 presence–absence grids, we align our setup with current large-scale benchmarks for species composition and presence prediction that explicitly target continental-scale, high-resolution community modelling (Picek et al., 2024; 2025). Together, the LatentMLP and ConvLSTM baselines span a representative spectrum from classical joint SDMs to modern spatio–temporal deep jSDMs, providing a strong baselines to compare BioAnalyst.

**Climate linear probing.**  To quantify how much climate-relevant information is encoded in our learned representations, we adopt a climate linear probing setup: the encoder is frozen and we fit a single linear layer to predict gridded climate variables or indices from the embeddings, following the idea of linear classifier probes as a measure of linear separability in deep features (Alain & Bengio,

2016). To contextualise these linear-probe scores against widely used non-parametric methods in climate science, we additionally train two baselines on the same features: a Random Forest (RF) and a Support Vector Machine (SVM). RFs have recently been shown to be highly effective for statistical downscaling and bias adjustment of climate simulations over Europe, for example through a posteriori random forests that stochastically downscale daily precipitation at dozens of European stations while accurately capturing local conditional distributions (Legasa et al., 2022). Complementarily, SVM-based approaches have been used at the global scale to post-process subseasonal-to-seasonal precipitation forecasts from the European Centre for Medium-Range Weather Forecasts, significantly improving skill over the raw dynamical model output (Yin et al., 2023). Reporting linear-probe performance alongside RF and SVM therefore allows us to disentangle how much climate signal is linearly accessible in the representation space of BioAnalyst versus what can be extracted by strong off-the-shelf non-linear baselines.

## C.5 METRICS

### C.5.1 PRE-TRAINING

**Mean Absolute Error (MAE):** To evaluate the performance of our model during pre-training, we measure and log the MAE between the predictions and the ground truth target which is

$$\text{MAE} = \frac{1}{V} \sum_{v=1}^{V} \frac{1}{H \times W} \sum_{i=1}^{H} \sum_{j=1}^{W} ||\hat{\mathbf{X}}_{i,j}^v - \mathbf{X}_{i,j}^v|| \tag{18}$$

where $i, j$ index over the longitude and latitude dimensions $H, W$ of each variable $v \in V$.

### C.5.2 TASK-SPECIFIC FINETUNING

**Root Mean Square Error (RMSE):** To evaluate the performance of the finetuned model in both biotic, abiotic tasks, we measure and log the RMSE between the predictions and the ground truth target which is

$$\text{RMSE} = \frac{1}{V} \sum_{v=1}^{V} \sqrt{\frac{1}{H \times W} \sum_{i=1}^{H} \sum_{j=1}^{W} (\hat{\mathbf{X}}_{i,j}^v - \mathbf{X}_{i,j}^v)^2} \tag{19}$$

**$F_1$ score:** To evaluate the performance of biotic fine-tuning task and obtain a comparison metric with the downstream dataset used.

$$F_1 = \frac{1}{N} \sum_{i=1}^{N} \frac{\text{TP}_i}{\text{TP}_i + \frac{\text{FP}_i + \text{FN}_i}{2}} \tag{20}$$

where 
$\text{TP}_i$ = number of correctly predicted species (true positives),

$\text{FP}_i$ = species predicted but *not* observed (false positives),

$\text{FN}_i$ = species present but *not* predicted (false negatives),

$N$ = number of evaluation units (e.g. sites or grid cells).

**Coefficient of determination:** To evaluate the performance of the abiotic linear probing task

$$R^2 = 1 - \frac{\sum_{i=1}^{n} (y_i - \hat{y}_i)^2}{\sum_{i=1}^{n} (y_i - \bar{y})^2}, \qquad \bar{y} = \frac{1}{n} \sum_{i=1}^{n} y_i, \tag{21}$$

where $y_i$ are the observed values, $\hat{y}_i$ are the predicted values, $\bar{y}$ is the sample mean of the observations and $n$ is the number of data points.

**Sørensen similarity map:** To compare the species sets in each grid-cell, using the incidence (presence/absence) form of the Sørensen–Dice coefficient

$$S_{ij} \;=\; \frac{2\,c_{ij}}{2\,c_{ij} + b_{ij} + d_{ij}}, \tag{22}$$

where $c_{ij}$ is the number of species present in both ground-truth and prediction at cell $(i,j)$, $b_{ij}$ those present only in the observation GBIF data, and $d_{ij}$ species only predicted by the model. The mean assemblage similarity estimate is an informative indicator for the performance of the species distribution model (SDM) (Pinto-Ledezma & Cavender-Bares, 2021).

Our community comparison uses an incidence-based Sørensen coefficient per grid cell, the similarity pattern captures a combination of (i) actual model skill and (ii) where the GBIF data are dense and reliable enough to evaluate species assemblages. In well-sampled regions, more species are recorded, detection probabilities are higher, and the model is trained on richer data; not surprisingly, these are also the areas where Sørensen similarity tends to be highest. In poorly sampled regions, many species are undetected, so unobserved but present species are treated as absences in the metric, which depresses Sørensen scores regardless of how good the underlying model might be. Assemblage-level SDM evaluations using Sørensen and related indices are known to be sensitive to such incompleteness. In addition, GBIF data are known to be geographically and taxonomically biased [1], with much denser sampling in Western and Central Europe, near cities and easily accessible areas, and substantially sparser coverage in Eastern and South-eastern Europe and remote regions. This pattern has been documented repeatedly for GBIF and similar occurrence databases (Beck et al., 2014).

## D  BIOANALYST ABLATIONS AND INTERPRETABILITY

In order to understand which input modalities contribute most to biodiversity predictions, we decided to observe the cross-attention weights from the encoder's first layer, which directly measures how the model allocates computational focus across input features before hierarchical processing in the Swin backbone. For a representative test window (December 2019), consider Figure 9, which shows that climate variables receive the highest attention, followed by species and surface observations, while most other modality groups receive lower attention. This ranking seems consistent across the 13 prediction windows in the test set Figure 10), with climate, species and surface maintaining high attention. Spatial analysis of the same representative window, particularly in the case of a high-interest variable (i.e., species), reveals that attention is not uniformly distributed: the encoder concentrates on Scandinavia and Central Europe regions Figure 12), which are hotspots where the strongest species occurrence signals would be encountered in the used data set. We focused on encoder cross-attention rather than decoder attention because it provides a direct, interpretable measure of which input modalities the model prioritises during initial feature extraction, before representations are hierarchically mixed by the Swin backbone, making modality contributions easier to quantify and attribute.

Additionally, Table Table 9 demonstrates the MAE performance across different variable groups, over the test set. The columns represent specific modality ablation studies: **C1** (Species $\times$ Climate), **C2** (Species $\times$ Edaphic), **C3** (Species $\times$ RLI), and **C4** (Species $\times$ Surface), compared against a more complex model **C5** (Species $\times$ Edaphic $\times$ RLI $\times$ Climate $\times$ Surface) and the **BFM Full** model combining all listed previously variables. Cells marked with "XXX" indicate variables that are out of scope for that specific modality combination. The results show that while the more comprehensive models (C5 and BFM Full) provide consistent predictions across physical variables (Surface, Edaphic, and Climate groups), single-modality combinations can offer slightly better performance in specific tasks. For example, for species abundance predictions (bottom section), the Climate-only model (**C1**) frequently outperforms the fuller models (e.g., for species ID 9809229, C1 achieves an MAE of 0.627 compared to 0.740 for C5). This could suggest that either climatic variables are the

---

[1] https://www.gbif.org/data-use/6hL40kh9ikDXftobIM85KP/sampling-biases-shape-our-view-of-the-natural-world

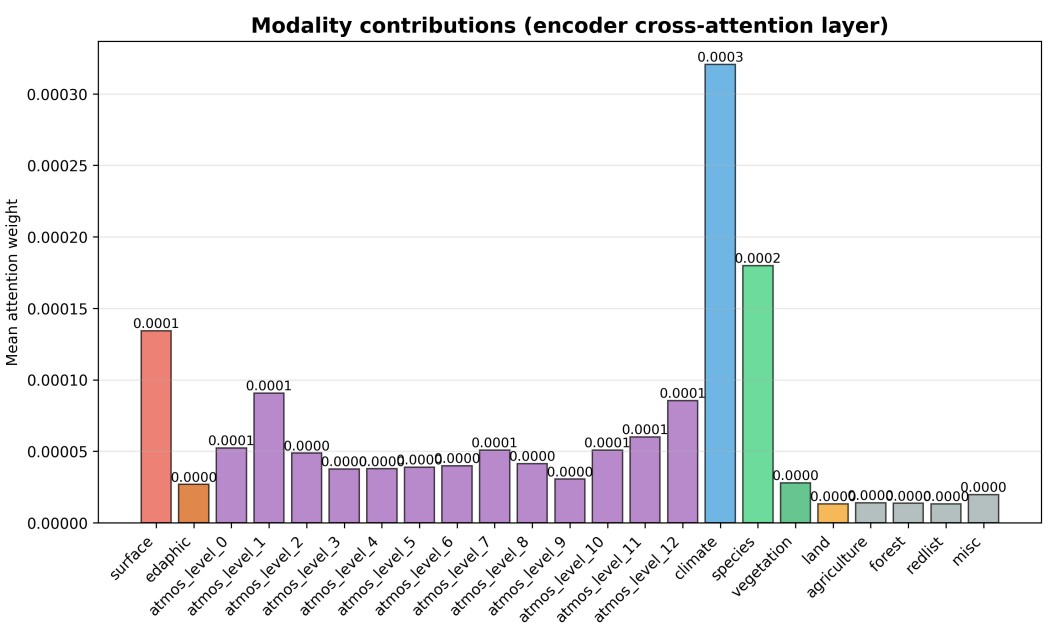

Figure 9: Mean attention weight per modality group contribution for the Encoders cross-attention layer.

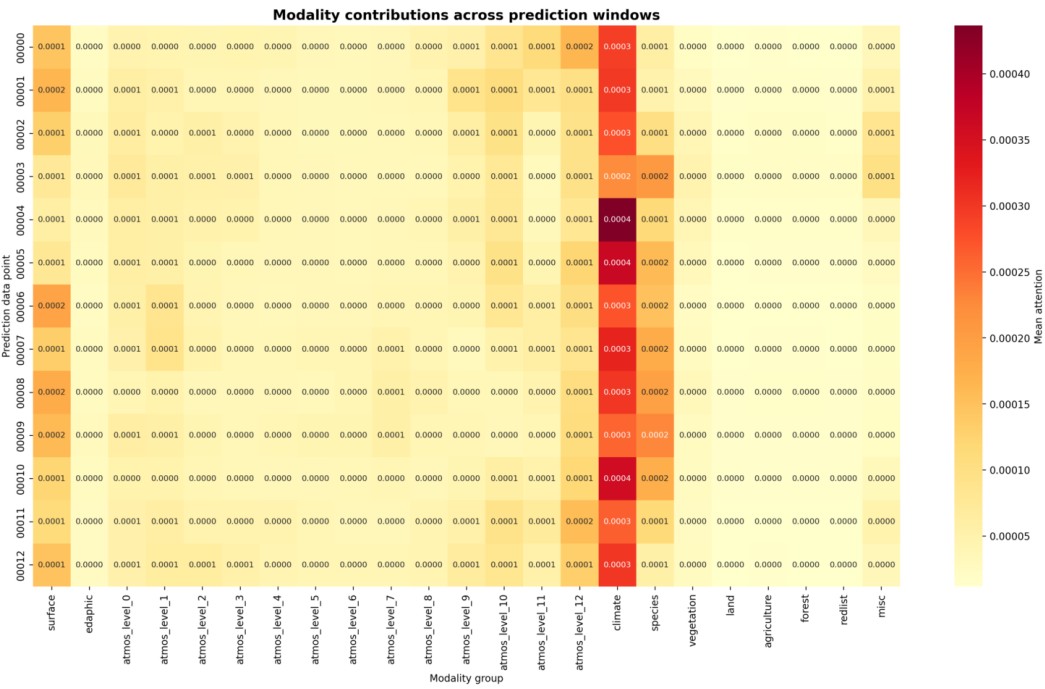

Figure 10: Mean attention weight per modality group contributions for 12 test windows (months).

dominant helpers in predicting species distributions in this dataset, and adding additional modalities (as in C5) may occasionally introduce noise rather than distinct predictive signals for certain species.

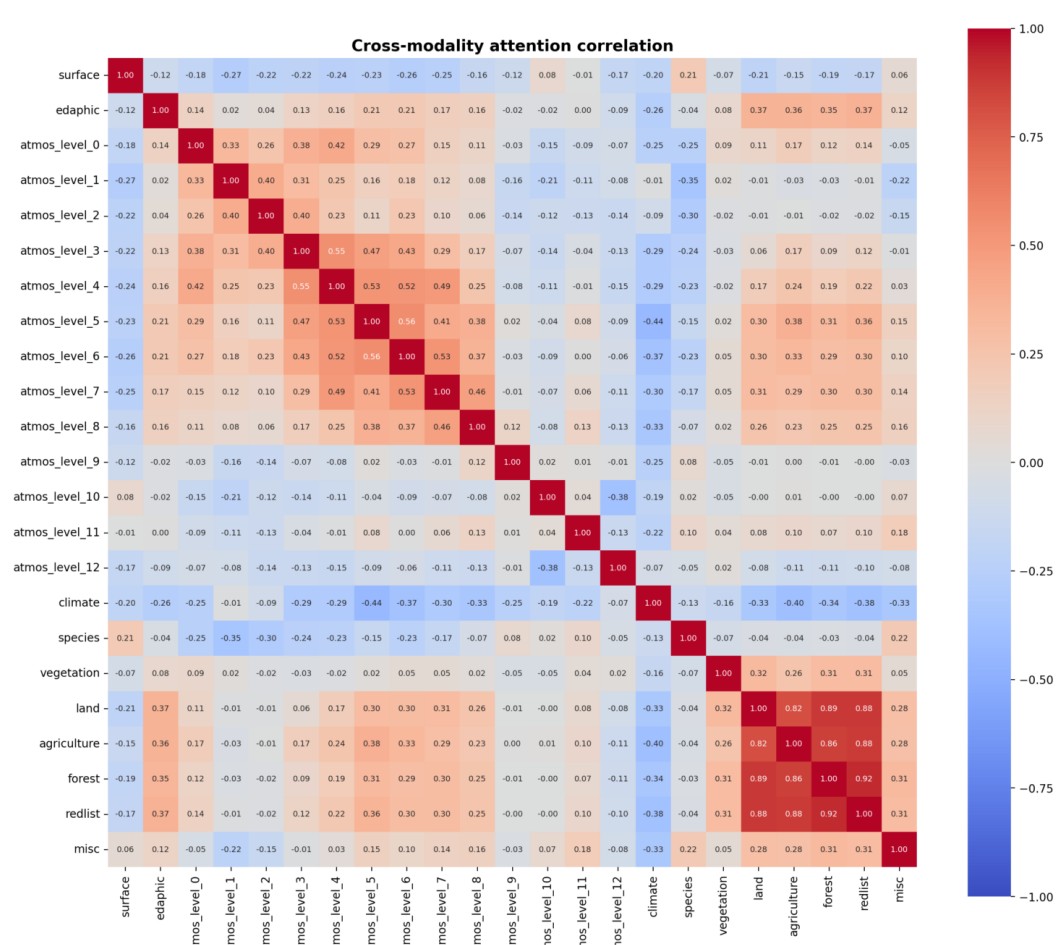

Figure 11: Cross-modality attention correlation matrix for all the modality groups with their respective variables.

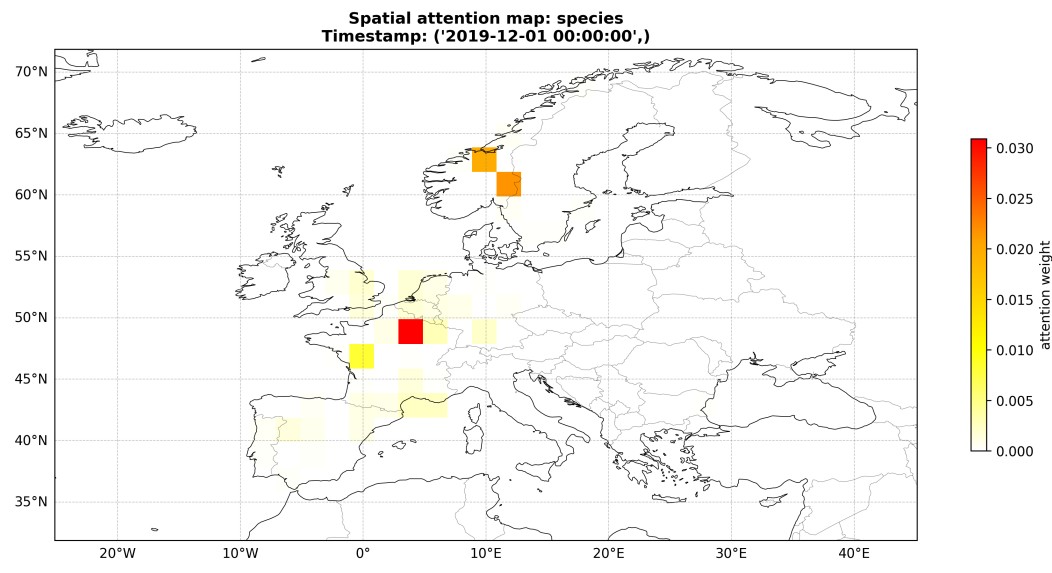

Figure 12: Cross-modality spatial attention map for the species variable group.

Table 9: Performance comparison between different modality combinations

| Group ($V$) | Var ($v$) | C1 | C2 | C3 | C4 | C5 | BFM Full |
|---|---|---|---|---|---|---|---|
| Surface | t2m | XXX | XXX | XXX | 2.753380 | 0.003875 | 3.007635 |
| | msl | XXX | XXX | XXX | 433.657866 | 421.848398 | 400.324847 |
| | slt | XXX | XXX | XXX | 0.005178 | 0.008991 | 0.013444 |
| | z | XXX | XXX | XXX | 16.460119 | 23.041222 | 33.079982 |
| | u10 | XXX | XXX | XXX | 1.344401 | 1.260686 | 1.218697 |
| | v10 | XXX | XXX | XXX | 1.156675 | 1.093683 | 1.105192 |
| | lsm | XXX | XXX | XXX | 0.002361 | 0.003875 | 0.004778 |
| Edaphic | swvl1 | XXX | 0.143530 | XXX | XXX | 0.015753 | 0.016012 |
| | swvl2 | XXX | 0.139144 | XXX | XXX | 0.016992 | 0.016745 |
| | stl1 | XXX | 6.284586 | XXX | XXX | 2.444646 | 2.544988 |
| | stl2 | XXX | 6.185118 | XXX | XXX | 2.403322 | 2.490814 |
| Atmospheric (p-levels) | z_pl | XXX | XXX | XXX | XXX | XXX | 77295.114183 |
| | t_pl | XXX | XXX | XXX | XXX | XXX | 39.698364 |
| | u_pl | XXX | XXX | XXX | XXX | XXX | 6.898406 |
| | v_pl | XXX | XXX | XXX | XXX | XXX | 1.645421 |
| | q_pl | 0.78 | XXX | XXX | XXX | XXX | 0.005081 |
| Climate | smlt | 0.000228 | XXX | XXX | XXX | 0.000116 | 0.000124 |
| | tp | 0.001278 | XXX | XXX | XXX | 0.000890 | 0.000857 |
| | csfr | 0.000001 | XXX | XXX | XXX | 0.000001 | 0.000001 |
| | avg_sdswrf | 80.703410 | XXX | XXX | XXX | 35.588897 | 34.508076 |
| | avg_snswrf | 71.948338 | XXX | XXX | XXX | 31.350053 | 30.553276 |
| | avg_snlwrf | 15.075831 | XXX | XXX | XXX | 7.237449 | 7.593424 |
| | avg_tprate | 0.000015 | XXX | XXX | XXX | 0.000010 | 0.000010 |
| | avg_sdswrfcs | 98.768588 | XXX | XXX | XXX | 44.961979 | 44.932251 |
| | sd | 0.060704 | XXX | XXX | XXX | 0.031556 | 0.018263 |
| | t2m_clim | 6.429481 | XXX | XXX | XXX | 2.730068 | 3.021820 |
| | d2m | 5.719590 | XXX | XXX | XXX | 2.675545 | 2.732889 |
| Vegetation | NDVI | XXX | XXX | XXX | XXX | XXX | 0.033334 |
| Land cover | Land | XXX | XXX | XXX | XXX | XXX | 982.858652 |
| Agriculture | Agriculture | XXX | XXX | XXX | XXX | XXX | 0.369489 |
| | Arable | XXX | XXX | XXX | XXX | XXX | 0.196134 |
| | Cropland | XXX | XXX | XXX | XXX | XXX | 0.035131 |
| Forest | Forest | XXX | XXX | XXX | XXX | XXX | 0.201663 |
| Redlist | RLI | XXX | XXX | 0.002770 | XXX | 0.003895 | 0.004125 |
| Miscellaneous | avg_slhtf | XXX | XXX | XXX | XXX | XXX | 5.487877 |
| | avg_pevr | XXX | XXX | XXX | XXX | XXX | 0.000004 |
| Species | 1340361 | 0.000042 | 0.000061 | 0.000042 | 0.000046 | 0.000043 | 0.000064 |
| | 1340503 | 0.063857 | 0.064053 | 0.063795 | 0.063897 | 0.063929 | 0.064850 |
| | 1536449 | 0.034173 | 0.037600 | 0.034158 | 0.034184 | 0.034193 | 0.034896 |
| | 1898286 | 0.322074 | 0.383199 | 0.335297 | 0.356860 | 0.358814 | 0.359776 |
| | 1920506 | 0.276737 | 0.267358 | 0.244421 | 0.264019 | 0.265405 | 0.272244 |
| | 2430567 | 0.001140 | 0.009217 | 0.001117 | 0.001168 | 0.001171 | 0.001872 |
| | 2431885 | 0.011713 | 0.012341 | 0.011626 | 0.011719 | 0.011766 | 0.012577 |
| | 2433433 | 0.000929 | 0.000936 | 0.000929 | 0.000934 | 0.000934 | 0.001001 |
| | 2434779 | 0.000014 | 0.000021 | 0.000014 | 0.000014 | 0.000014 | 0.000018 |
| | 2435240 | 0.004547 | 0.007200 | 0.004532 | 0.004726 | 0.004585 | 0.005042 |
| | 2435261 | 0.000050 | 0.000066 | 0.000049 | 0.000052 | 0.000052 | 0.000074 |
| | 2437394 | 0.000181 | 0.000183 | 0.000180 | 0.000183 | 0.000184 | 0.000220 |
| | 2441454 | 0.004198 | 0.004282 | 0.004192 | 0.004248 | 0.004272 | 0.005205 |
| | 2473958 | 0.049975 | 0.046870 | 0.049680 | 0.049740 | 0.049872 | 0.051274 |
| | 2491534 | 0.246025 | 0.252939 | 0.267842 | 0.282393 | 0.284369 | 0.294113 |
| | 2891770 | 0.043579 | 0.043935 | 0.043551 | 0.043735 | 0.043834 | 0.046475 |
| | 3034825 | 0.021265 | 0.021283 | 0.021218 | 0.021394 | 0.021357 | 0.023135 |
| | 4408498 | 0.015352 | 0.015404 | 0.015298 | 0.015445 | 0.015424 | 0.016373 |

**Table 9 – continued from previous page**

| Group ($V$) | Var ($v$) | C1 | C2 | C3 | C4 | C5 | BFM Full |
|---|---|---|---|---|---|---|---|
| | 5218786 | 0.007418 | 0.007418 | 0.007412 | 0.007432 | 0.007430 | 0.007713 |
| | 5219073 | 0.000768 | 0.000766 | 0.000763 | 0.000770 | 0.000769 | 0.000831 |
| | 5219173 | 0.006217 | 0.006225 | 0.006212 | 0.006230 | 0.006233 | 0.006481 |
| | 5219219 | 0.000254 | 0.000255 | 0.000254 | 0.000261 | 0.000256 | 0.000296 |
| | 5844449 | 0.006492 | 0.006508 | 0.006476 | 0.006517 | 0.006520 | 0.006944 |
| | 8002952 | 0.015073 | 0.015116 | 0.015069 | 0.015131 | 0.015135 | 0.015838 |
| | 8077224 | 0.318140 | 0.329542 | 0.340779 | 0.359972 | 0.357112 | 0.361090 |
| | 8894817 | 0.000202 | 0.000228 | 0.000199 | 0.000217 | 0.000208 | 0.000336 |
| | 8909809 | 0.005668 | 0.005821 | 0.005664 | 0.005761 | 0.005711 | 0.006676 |
| | 9809229 | 0.627497 | 0.646721 | 0.719715 | 0.735913 | 0.739663 | 0.745430 |

