# OpenReview forum: "BioAnalyst: A Foundation Model for Biodiversity"
_ICLR.cc/2026/Conference — Submitted to ICLR 2026_

### Official Review · Reviewer_9YE4 · 2025-10-24

**Soundness:** 1
**Presentation:** 3
**Contribution:** 2
**Rating:** 2
**Confidence:** 3

**Summary:**

In this paper the authors propose BioAnalyst a  model presented as a forecasting foundation model for biodiversity in Europe at 0.25 degrees resolution.  More specifically, the model is trained with BioCube which encompasses 10 modalities including climate and species range data. The proposed architecture is a composed of a PerceiverIO encoder, a 3d swin transformer backbone and a PerceiverIO decoder.
The model is  pre-trained on the task of forecasting at the next time step. Then the model is evaluated on downstream tasks including forecasting biodiversity dynamics, and species distributions.

**Strengths:**

This project tackles an important challenge and few studies have incorporated these many types of input for biodiversity monitoring applications. The paper is generally well structured too.

**Weaknesses:**

In general, the applications in ecology would benefit from being explained in a clearer way:

- For example, the task could be better defined especially when referring to species distribution: there are many things that can be modelled in species distribution modelling (encounter rates, abundance, occurrence, occupancy). It is unclear what you are trying to model. Can you please clarify what is predicted in terms of species distribution?

- I would like to understand how you think that a 28km resolution is what is needed for conservation decisions. I would perhaps be a bit more conservative about this statement.  It would be good for the authors to outline more precisely what kind of decisions a map at this scale could be helpful for. In practice for conservation decisions especially when looking at specific species (which is the use case the authors are highlighting), much higher resolution is needed.  I am also surprised this is the resolution that is chosen given the higher resolution of many of the products that are used (e.g. satellite imagery) especially in Europe. It seems like this alignment at 28km loses a lot of the fine-grainedness of the original raw data source. I understand that this might be the approach taken with BioCube but I wonder if then it is not the most suited dataset for this task.

Evaluation:
- I am confused about the choice of looking at cumulative mean of distribution across 28 species distributions (Fig 3). This actually doesn't reflect error. Instead you should show the error (you can average it across space and species if you want) but otherwise it can be very misleading. also, the claim "The predictions closely follow the data trend, " doesn't seem very well supported.

**Questions:**

In addition to asking for clarification about the points in the weaknesses, I have the following questions:

- the model is pre-trained with prediction one time step ahead from the past. Hae the authors tried other number of steps ? Or random number of steps (with additional conditioning on the number of steps to predict ahead) ? Why would a task of interpolation not be better suited?  How well is the model able to predict on longer time frames?

Evaluation:
- sorensen similarity map in Fig 4: I almost seems like the species match is correlated with data sampling (if we think about number of observations in general). Could the authors elaborate on that? Are there spatial biases in the species data that is used to validate these maps?
- Fig 6: I might have misunderstood but it seems that in the ground truth not all data is observed. How do you know that Aurora overpredicts if there are only a few cells with which to validate the map?

- On another note, this is  more of a comment about something that could be highlighted in the discussion than a weakness, but the RLI is not necessarily suited for this task. I would be curious to see if there are ablations of the different modalities used for this model. Indeed, the problem with the IUCN red list is that for a given species, its assessment is updated every 10 years.

- predicting at the next time step: how do the authors think this can be suited for capturing things like seasonal patterns?
-  C.4.2.: I think this would benefiot from clarification about how the observations are aggreggated, and how "The model preserves the full spatial resolution of the data" since the observations of geolifeclef are point data?

Other comments: please cite the references with parentheses, it makes it difficult to parse the sentences otherwise.

I am open to revising my score, but in the current state, the definition of the downstream tasks is quite unclear to me as well as the pathway to impact.

---

> ### Author Response · Authors · 2025-11-19
> **Reply to weaknesses of Reviewer 9YE4 Part 1/3**
>
> We would like to thank the reviewer for his time spend on reading our manuscript and providing us with improvement cues which we have followed. Below we reply to both weaknesses and questions one by one on parts.
>
> ### Weakness 1: For example, the task could be better defined especially when referring to species distribution: there are many things that can be modelled in species distribution modelling (encounter rates, abundance, occurrence, occupancy). It is unclear what you are trying to model. Can you please clarify what is predicted in terms of species distribution?
>
> In the biotic downstream task, our objective is to model species occurrences rather than encounter rates, occupancy, or abundance in the formal ecological sense. Specifically, we use the GeoLifeCLEF 2024 dataset and predict presence-absence occurrences for 500 species across multiple years, aggregated spatially to a common grid. To obtain a consistent spatial target, all observation records for a species within a grid cell and year are aggregated into a binary occurrence label (presence if at least one observation is recorded; absence otherwise). In the post-processing phase, we aggregate statistics as species abundance per grid cell for visualisation (Figure 6 in paper). Although this results in a presence–absence modelling setup, the multi-taxon nature of the dataset means we effectively perform joint species distribution modelling (jSDM) across space and time. jSDMs are notoriously difficult to scale using classical/Bayesian statistics. We have updated the manuscript to clearly describe what is being predicted and how these labels are constructed, plus added more references to literature highlighting the nature of the task.
>
> Extra references:
>
> König, Christian, et al. "Scale dependency of joint species distribution models challenges interpretation of biotic interactions." Journal of Biogeography 48.7 (2021): 1541-1551https://doi.org/10.1111/jbi.14106 konig2021scale
>
> Tikhonov, G., Opedal, Ø. H., Abrego, N., Lehikoinen, A., de Jonge, M. M., Oksanen, J., & Ovaskainen, O. (2020). Joint species distribution modelling with the R‐package Hmsc. Methods in ecology and evolution, 11(3), 442-447. https://doi.org/10.1111/2041-210X.13345 tikhonov2020joint

---

> ### Author Response · Authors · 2025-11-19
> **Reply to weaknesses of Reviewer 9YE4 Part 2/3**
>
> ### Weakness 2: I would like to understand how you think that a 28km resolution is what is needed for conservation decisions. I would perhaps be a bit more conservative about this statement. It would be good for the authors to outline more precisely what kind of decisions a map at this scale could be helpful for. In practice for conservation decisions especially when looking at specific species (which is the use case the authors are highlighting), much higher resolution is needed. I am also surprised this is the resolution that is chosen given the higher resolution of many of the products that are used (e.g. satellite imagery) especially in Europe. It seems like this alignment at 28km loses a lot of the fine-grainedness of the original raw data source. I understand that this might be the approach taken with BioCube but I wonder if then it is not the most cont...
>
> Thank you for this comment, which is similar to reviewer 2dxb27: Enhancing that we give a more complete response below:
> We agree that many site–level conservation decisions for individual species (e.g. designing a reserve boundary, restoring a specific wetland) require much finer spatial resolution than 28 km and often benefit from micro-climatic or habitat information at tens–hundreds of metres. Our intention was not to claim that a 0.25° grid is sufficient for all conservation applications, but rather to target regional to national–scale questions where this resolution is commonly used.
> First, one of BioAnalyst's targets is to act as a large-scale joint species distribution model. JSDMs are increasingly used to analyse community composition and drivers over broad extents in macroecology and conservation, but their computational cost and data requirements make truly high-resolution, continental-scale applications challenging. A 0.25° (~28 km) grid is a pragmatic compromise that allows us to model multi-species dynamics across Europe while keeping training tractable.
> Second, our primary climatic driver is ERA5, whose standard global product is provided on a 0.25° grid with high temporal resolution and long temporal coverage. Aligning BioAnalyst to this grid enables direct comparison with climate-based FMs like Aurora (which is also trained on ERA5). Although higher-resolution derivatives such as ERA5-Land exist, many of our other modalities (species,vegetation, agriculture) are not consistently available at those finer resolutions across the full 2000–2020 period.
> Third, there is a substantial body of work in macroecology and conservation planning that operates at comparable or even coarser resolutions when addressing broad-scale questions (e.g. global biodiversity risk, identification of priority regions, assessment of conservation networks) [Engemann et al. 2015 ] . For example, recent global analyses of climate-change risks to terrestrial biodiversity and macroecological diversity patterns have used 25–50 km grid cells, arguing that such coarse units are appropriate for capturing broad climatic and biogeographic gradients relevant to policy [Roll et al. 2017]. Similarly, assessments of Key Biodiversity Areas and other large-scale conservation prioritization exercises often use 25–50 km planning units. Thus, a 0.25° grid is well aligned with the spatial grain at which many national or continental assessments are currently performed [Farooq et al. 2023]. We have updated our manuscript on Appendix section B.1 with this extra information.
> We have revised the manuscript to clarify which kinds of decisions BioAnalyst is intended to support starting from our abstract. Specifically, on section 3 - Method we now state that the 28 km resolution is appropriate for regional to national-scale applications such as (i) mapping broad richness and community-composition patterns, (ii) identifying large-scale hotspots and coldspots under different climate scenarios, and (iii) informing high-level prioritization or reporting (e.g. national assessments, EU-wide strategies). We explicitly acknowledge in our conclusion and discussion Section 5 that local habitat management and single-site reserve design will require downscaling or complementary, higher-resolution models, and that BioAnalyst should be seen as a macroecological simulator rather than a replacement for fine-grain SDMs.
> Refs:
>
> [1] Farooq, Harith, Alexandre Antonelli, and Søren Faurby. "A call for improving the Key Biodiversity Areas framework." Perspectives in Ecology and Conservation 21.1 (2023): 85-91.
>
> [2] Roll, U., Feldman, A., Novosolov, M., Allison, A., Bauer, A. M., Bernard, R., ... & Meiri, S. (2017). The global distribution of tetrapods reveals a need for targeted reptile conservation. Nature ecology & evolution, 1(11), 1677-1682.
>
> [3] Engemann, K., Enquist, B. J., Sandel, B., Boyle, B., Jørgensen, P. M., Morueta‐Holme, N., ... & Svenning, J. C. (2015). Limited sampling hampers “big data” estimation of species richness in a tropical biodiversity hotspot. Ecology and evolution, 5(3), 807-820.

---

> ### Author Response · Authors · 2025-11-19
> **Reply to weaknesses of Reviewer 9YE4 Part 3/3**
>
> ### Weakness 3: I am confused about the choice of looking at cumulative mean of distribution across 28 species distributions (Fig 3). This actually doesn't reflect error. Instead you should show the error (you can average it across space and species if you want) but otherwise it can be very misleading. also, the claim "The predictions closely follow the data trend, " doesn't seem very well supported.
>
> We would like to thank the reviewer for the constructive comment. Following their suggestion, we have adjusted Figured 3. and created a new visualisation highlighting the MAE across species over time for 2 different rollout finetuning horizons. More specific, now our text reads:
> “First we present the MAE for the species variable group (28 animal species), comparing the performance of a 12-step roll-out for our mix horizons $K=6$ and $K=12$ time-steps respectively \autoref{fig:species_cumulative}. The 12-step finetuned variant follows a lower MAE across the 12-step trajectory, validating the effectiveness of our mix horizon scheme during roll-out finetuning. Both variants exhibit increasing MAE after 10 steps.”

---

> ### Author Response · Authors · 2025-11-19
> **Reply to questions of Reviewer 9YE4 Part 1/4**
>
> ### Question 1: the model is pre-trained with prediction one time step ahead from the past. Hae the authors tried other number of steps ? Or random number of steps (with additional conditioning on the number of steps to predict ahead) ? Why would a task of interpolation not be better suited? How well is the model able to predict on longer time frames?
>
> Thank you for pointing that out. The same remark was put forward from  Reviewer gCqn and we have made it more explicit in our text on section 3.4 “Finetuning” under bolded sections “Short-lead time finetuning” and “Rollout finetuning”. Also we have added a paragraph with more details on Appendix C.4.1 which reads:  BioAnalyst has been pre-trained to predict $x_{t+1}$ by providing as input $[x_{t-1}, x_{t}]$. During fine-tuning and more specific, roll-out fine-tuning, BioAnalyst was extended to predict 6 and 12 steps ahead in 2 stages. At the first stage, we fine-tune the complete BioAnalyst, adding VeRA adapters on the backbone for 12000 steps with a horizon of 6 time-steps (6 months), using constant learning rate of $0.00003$, weight decay of $0.000003$ and store the weights. In the second stage we load the previously stored weights and continue the roll-out fine-tuning for 12000 time-steps more, with a horizon of 12 steps (1 year), using constant learning rate of $0.00001$ and weight decay of $0.000001$. Fine-tuning with a mix of variable horizons is a technique used on Aurora \citep{bodnar2024aurora}, PrithviWxC \citep{schmude2024prithviwxcfoundationmodel} and Aardvak \citep{allen2025end}, offering improved long-horizon forecasting capabilities, which we also observed ourselves and are highlighted on Figure 9 for the species variables.
> Regarding longer time frames predictions, we found that BioAnalyst performs very well up-until 10 timesteps ahead. After that, the error increases on our tests for 12 timesteps as depicted on Figure 3.

---

> ### Author Response · Authors · 2025-11-19
> **Reply to questions of Reviewer 9YE4 Part 2/4**
>
> Question 2: sorensen similarity map in Fig 4: I almost seems like the species match is correlated with data sampling (if we think about number of observations in general). Could the authors elaborate on that? Are there spatial biases in the species data that is used to validate these maps?
>
> We thank the reviewer for highlighting this important point.
>
> Yes, there are strong spatial biases in the species data used for this analysis, and this is reflected in the Sørensen map. In this part, the results are coming from pre-training and fine-tuning on BioCube which uses GBIF animal species occurrence data (presence-only records). GBIF data are known to be geographically and taxonomically biased, with much denser sampling in Western and Central Europe, near cities and easily accessible areas, and substantially sparser coverage in Eastern and South-eastern Europe and remote regions. This pattern has been documented repeatedly for GBIF and similar occurrence databases [1], [2]. Because our community comparison uses an incidence-based Sørensen coefficient per grid cell, the similarity pattern captures a combination of (i) actual model skill and (ii) where the GBIF data are dense and reliable enough to evaluate species assemblages. In well-sampled regions, more species are recorded, detection probabilities are higher, and the model is trained on richer data; not surprisingly, these are also the areas where Sørensen similarity tends to be highest. In poorly sampled regions, many species are undetected, so unobserved but present species are treated as absences in the metric, which depresses Sørensen scores regardless of how good the underlying environmental model might be. Assemblage-level SDM evaluations using Sørensen and related indices are known to be sensitive to such incompleteness. To make this clearer, we have revised the text in the Results to explicitly acknowledge that the Sørensen map should be interpreted as the interplay of model performance and spatially uneven sampling effort, and that our mean value of 0.31 reflects moderate compositional skill in the better-sampled parts of Europe, rather than uniform performance everywhere. Now our text reads:  “Second, we report a mean Sørensen similarity of 0.31 for 28 animal species across all evaluated land grid cells, indicating that the predicted assemblages recover roughly one-third of the observed community composition. \autoref{fig:sorensen} highlights spatial variation in assemblage agreement, with higher similarity in western and central Europe and lower values in eastern and south-eastern regions. This pattern closely follows known spatial biases in the underlying GBIF occurrence data, with denser sampling in some regions than others, so the map reflects both model skill and uneven observation effort. Overall, the spatial pattern suggests moderate compositional skill in well-sampled areas, useful for broad biogeographic inference, yet leaving room for improving species co-occurrence dynamics.”
>
> We also revised the figure caption (see below) to (i) explicitly mention that the validation data come from GBIF, (ii) note the known spatial sampling bias, and (iii) clarify that warm colours highlight cells where predicted and recorded assemblages match reasonably well, whereas cool colours include both model error and lack of records.
>
> Finally, we corrected a typo in the Sørensen formula: in the denominator, species present only in the prediction are now denoted $d_{ij}$, so that:
>
> \begin{equation}
>     S_{ij}\;=\;\frac{2\,c_{ij}}{2\,c_{ij}+b_{ij}+d_{ij}},
> \end{equation}
>
> with $c_{ij}$ the number of shared species, species only observed in GBIF, and species only predicted by the model.
>
> References:
>
> [1] https://www.gbif.org/data-use/6hL40kh9ikDXftobIM85KP/sampling-biases-shape-our-view-of-the-natural-world
>
> [2] Beck, Jan, et al. "Spatial bias in the GBIF database and its effect on modeling species' geographic distributions." Ecological Informatics 19 (2014): 10-15.

---

> ### Author Response · Authors · 2025-11-19
> **Reply to questions of Reviewer 9YE4 Part 3/4**
>
> Question 3: Fig 6: I might have misunderstood but it seems that in the ground truth not all data is observed. How do you know that Aurora overpredicts if there are only a few cells with which to validate the map?
>
> We thank the reviewer for raising this important point. We use from GeoLifeCLEF the presence-absenc part, where ground-truth data cover all grid cells, and that blank cells in panel (a) represent true zeros, absence of species occurrence.
> Our use of ‘overestimation’ was intended to be qualitative, referring to the visibly higher and more spatially extensive richness values produced by Aurora relative to the observed footprint. We recognise that, given the spatial coverage of the ground truth, this wording is too strong and potentially misleading. We have therefore toned down the claim and now explicitly describe Aurora as ‘produces a more spatially extensive richness field, assigning moderate richness also to cells without recorded observations’.
> We have also edited the Figure 6 caption such that it makes clear the white cells represent. Now it reads:
>  “Comparison of observed and predicted species richness in the Netherlands for the year 2021. a) The Observed Richness based on empirical field data from GeoLifeCLEF2024 presence-absence survey; blank cells represent true zeros indicating absence of species occurrence. b) The species richness for 2021 predicted by the BioAnalyst, and c) Aurora-0.25 predictions for 2021.”
> More specific, our text now reads:
> “Nevertheless, investigating the predicted species richness in more detail for both models, we observe that Aurora produces a more spatially extensive richness field, assigning moderate richness also to cells with recorded absent observations, which likely reflects extrapolation from smooth climate fields. BioAnalyst’s predictions remain more tightly constrained to the observed footprint, consistent with the additional process-related modalities it uses (land use, vegetation, species signals) \autoref{fig:species_richness}. This result supports the view that, for complex community-level processes, climate-only representations can be limiting, and that integrating additional process-related modalities, as in BioAnalyst, can substantially improve predictive”
>
>
> ### Question 4: On another note, this is more of a comment about something that could be highlighted in the discussion than a weakness, but the RLI is not necessarily suited for this task. I would be curious to see if there are ablations of the different modalities used for this model. Indeed, the problem with the IUCN red list is that for a given species, its assessment is updated every 10 years.
> Thank you for pointing this out and giving us the opportunity to shed some light to the RLI variable. We have made extensive experiments to uncover BioAnalysts inner modality representations capabilities based on yours and rest reviewers comments and remarks. For that reason, we have drafted a new section D. - BioAnalysts Ablations and interpretability with 4 new figures that highlight relevant information. More specifically, both Figure 9 and Figure 10 show that RLI is contributing near 0 to the cross-attention weights. These findings  validate your remark that this variable is not well suited for our tasks.
>
> ### Question 5: predicting at the next time step: how do the authors think this can be suited for capturing things like seasonal patterns?
>
> BioAnalyst is pre-trained on single time-step forecasting but is fine-tuned to 6 and 12-timesteps (1 year) forecast horizon, enabling multi-seasonal pattern representation. The final model is capable of a 12-step prediction window. This is also reflected on the caption of Figure 1, (iii). Also, we have updated our text on “Method” to clarify that early on. Now it reads: BioAnalyst can be thought of as a \textit{forecast emulator}, i.e., given a state of the Earth's biodiversity at times $t$ and $t-\delta t$, it predicts the state at $t + \delta t$, where $\delta$ is a discrete time step and the final fine-tuned model can predict up to 12 time-steps ahead.

---

> ### Author Response · Authors · 2025-11-19
> **Reply to questions of Reviewer 9YE4 Part 4/4**
>
> Question 6: C.4.2.: I think this would benefiot from clarification about how the observations are aggreggated, and how "The model preserves the full spatial resolution of the data" since the observations of geolifeclef are point data?
>
> We thank the reviewer for highlighting the need for clarification regarding the aggregation of point observations in GeoLifeCLEF and the statement about spatial resolution. GeoLifeCLEF provides point-level species observations, but for modelling purposes we aggregate these to a fixed spatial grid (matching the resolution used in the abiotic task). Each grid cell receives a binary presence label per species if at least one observation falls within that cell for a given year. Thus, we retain the full resolution of the prediction grid, not the raw point coordinates. The model outputs species occurrence predictions at every grid cell, preserving spatial detail in the final prediction map even though the raw input is point-based. We have revised the text to clarify both the aggregation procedure and what is meant by “preserving spatial resolution.”
>
>
> Other comments: please cite the references with parentheses, it makes it difficult to parse the sentences otherwise. :
> Thank you for spotting that, it has been fixed.

---

> ### Author Response · Authors · 2025-11-26
> **Rebuttal period is concluding soon**
>
> Dear reviewer 9YE4,
>
> Thank you again for the thoughtful feedback and constructive comments. As the rebuttal deadline is approaching, we would be grateful if you could take a look at our rebuttal and the updated text to ensure we have addressed your concerns accurately. If you have additional remarks, questions or requests, we are more than happy to address them! We appreciate your time and consideration.

---

### Official Review · Reviewer_2dxb · 2025-10-27

**Soundness:** 1
**Presentation:** 1
**Contribution:** 1
**Rating:** 2
**Confidence:** 3

**Summary:**

This article describe a foundation model pre-trained on the BioCube dataset. This dataset aggregates a diverse set of data sources into a geospatial data cube at a 0.25 deg resolution. These data sources cover different aspects of the environment that are relevant to biodiversity, including soil and climate variables or species observations.
The model is based on a PerceiverIO architecture for the encoder and decoder with a Swin Transformer as backbone, and the pre-training objective is a MAE.
Two downstream tasks are used for evaluation: species distribution mapping and forecasting based on GeoLifeCLEF2024 and monthly climate recovery based on CHELSA.

**Strengths:**

- The motivation is clear: there is currently no foundation model that has been pretrained on such a variety of variables relevant to biodiversity-related tasks.

**Weaknesses:**

1. I find the evaluation to be far from enough to drive the message of the paper. With only two tasks, one of them not directly related to biodiversity, it is hard to convince the readers that the proposed model actually serves as a foundation model. In addition, only one competing method is shown: Aurora, which is intended as a climate foundation model and is thus not a great fit for the species distribution task (I couldn’t find a comparison for the climate recovery task, where Aurora would probably do a good job). I would expect a comparison against several purely supervised baselines (with the same architecture and also with simpler architectures) and against more suitable competitors, including Earth observation-focused foundation models and pre-computed embeddings (such as AlphaEarth).
2. Even after having a look at the suppl. material and the BioCube paper, I struggle to understand the nature of the dataset used. For instance, Table 2 displays a list of species and their total number of occurrences. Does this mean that only occurrences of these species are used as input? Are these counts simply aggregated over 0.25 deg grid? In Eq. (4) we see how the input to the model is put together, but I find it confusing. Do all variables have 10 levels? (not just pressure?). If there are two nested sums, this means all variables and level are summed together? I have spent some time on this (again, both on this paper and the original BioCube), and I still don’t see how exactly the input data looks like.
3. I am unable to understand the nature of the used model. The description clearly states that encoder and decoder are based on cross-attention (as done in PerciverIO), and that, in between, there is a Swin Transformer. However, Swin Transformers are specifically designed for spatially arranged representations (such as image tokens), while the representation provided by a PerceiverIO encoder lacks any notion of spatial arrangement. How are both put together?
4. In connection to point 1, I felt there is a lack of discussion of what types of downstream tasks the model would be a good fit for. After all, the spatial resolution would make it unsuitable for many local downstream applications.

**Questions:**

I would like to read an answer to the questions formulated in weakness 2 and 3:
- What is the dimensionality of each data modality? How are the different modalities built?
- How are the PerceiverIO and Swin Transformer architectures put together? (given that the first provides non-spatial representations and the second requires explicitly spatial ones).
Other than this, I believe there is still a lot to do wrt weakness 1 in order to sustain the claim that the proposed model is a foundation model.

---

> ### Author Response · Authors · 2025-11-19
> **Reply to weaknesses of Reviewer 2dxb Part 1/2**
>
> We would like to thank the reviewer for the constructive criticism and extended remarks that further improved our manuscripts story and improve important areas. We will reply to both weaknesses and questions one by one below.
> ### Weakness 1: I find the evaluation to be far from enough to drive the message of the paper. With only two tasks, one of them not directly related to biodiversity, it is hard to convince the readers that the proposed model actually serves as a foundation model. In addition, only one competing method is shown: Aurora, which is intended as a climate foundation model and is thus not a great fit for the species distribution task (I couldn’t find a comparison for the climate recovery task, where Aurora would probably do a good job). I would expect a comparison against several purely supervised baselines (with the same architecture and also with simpler architectures) and against more suitable competitors, including Earth observation-focused foundation models and pre-computed embeddings (such as AlphaEarth)
> We appreciate the concern regarding the breadth of downstream tasks and the comparative baselines presented. It is important to emphasise that while one of our tasks (the abiotic climate‐probing) may appear less directly tied to biodiversity modelling, it is in fact fundamentally relevant: ecological niches and species‐distributions are strongly shaped by climatic conditions (temperature, precipitation, seasonality) and thus modelling these abiotic drivers is an integral part of biodiversity forecasting. For example, studies on ecological niche modelling repeatedly show that climatic variables are among the primary predictors of species presence and range boundaries. We added references to relevant literature in the paper for highlighting this [1].
> As climate is crucial and no similar models trained with the exact same modalities as BioAnaylst exist, we thought only a climate Foundation Model (i.e. Aurora) is a valid comparison with BioAnalyst as it captures the variation in ecological niches, especially since Aurora and BFM both used ERA-5 data for climate modality during pretraining. AlphaEarth (and other GeoFMs) was trained on remote sensing data based on multispectral optical instruments. A direct comparison cannot be made as the model represents very different physical processes.
> The manuscript  text has been updated to make this crucial detail more visible to qualify our choice for the Aurora comparison and the two downstream tasks.
>
> ### Weakness 2: Even after having a look at the suppl. material and the BioCube paper, I struggle to understand the nature of the dataset used. For instance, Table 2 displays a list of species and their total number of occurrences. Does this mean that only occurrences of these species are used as input? Are these counts simply aggregated over 0.25 deg grid? In Eq. (4) we see how the input to the model is put together, but I find it confusing. Do all variables have 10 levels? (not just pressure?). If there are two nested sums, this means all variables and level are summed together? I have spent some time on this (again, both on this paper and the original BioCube), and I still don’t see how exactly the input data looks like.
> We provide a complete answer to this weakness as a response to Question 1 which is related.
>
>
> Reference:
>
> [1] Helaouët, Pierre, and Gregory Beaugrand. "Physiology, ecological niches and species distribution." Ecosystems 12.8 (2009): 1235-1245.

---

> ### Author Response · Authors · 2025-11-19
> **Reply to weaknesses of Reviewer 2dxb Part 2/2**
>
> ### Weakness 3: I am unable to understand the nature of the used model. The description clearly states that encoder and decoder are based on cross-attention (as done in PerciverIO), and that, in between, there is a Swin Transformer. However, Swin Transformers are specifically designed for spatially arranged representations (such as image tokens), while the representation provided by a PerceiverIO encoder lacks any notion of spatial arrangement. How are both put together?
>
> This integration is achieved through a simple reshape operation that reinterprets the latent sequence as a 3D spatial volume. The Perceiver IO encoder outputs a flat sequence of latent tokens, which is then reshaped into a spatial tensor with dimensions (depth, height, width), where the depth dimensions naturally aligns with the number of modality groups and the spatial dimensions correspond to the patch grid. This creates a multi-channel feature map, similarly to standard computer vision inputs, which the Swin backbone can then process through hierarchical window-based attention on the spatial dimensions while treating depth as feature channels.
> After Swin processing, the inverse transformation flattens the spatial volume back into a sequential format for the Perceiver IO decoder, which uses cross-attention to generate pixel-level predictions. This design combines Perceiver IO’s strength at flexible, modality-agnostic encoding with Swin’s hierarchical processing. The depth alignment with modality groups is by design: the encoder initializes separate latent parameter sets for each modality group (each containing one latent per spatial patch), which are concatenated in a structured order before Perceiver IO processing, which ensures that the subsequent, deterministic reshaping will map each modality to its own depth slice in the 3D volume.
> We have updated our manuscript, and more specific Appendix A.2 - Design Choices to reflect that.
>
> ### Weakness 4: In connection to point 1, I felt there is a lack of discussion of what types of downstream tasks the model would be a good fit for. After all, the spatial resolution would make it unsuitable for many local downstream applications.
>
> Thank you for this observation which is also very closely related with a weakness observed by reviewer 9YE4. A first part of our answer is given here, but we recommend the interested reviewer to have a look into our extended response on reviewer’s 9YE4 remark further below.
> BioAnalyst is regional to national scale FM. We deliberately focused on this scale for 3 reasons:
> Large spatial scale joint species distribution modelling is a challenging task in Ecological modelling [König et al 2021], we want to address this gap with BioAnalyst.
> We had to choose between ERA5 and other climate datasets, ERA5 offers very high temporal resolution but at the cost of a coarser spatial resolution. Using ERA5 also makes the comparison to Aurora more plausible as Aurora also uses ERA5 as one of the pretraining modalities.
> Although ERA5 comes with higher resolutions, other modalities like agriculture, vegetation, species, do not. We came across cases that that the higher resolution has available but not for the time range we were looking for.
> We have revised the manuscript to clarify which downstream tasks the model would be a good fit based on our selected resolution. BioAnalyst is intended to support starting from our abstract. Specifically, on section 3 - Method we now state that the 28 km resolution is appropriate for regional to national-scale applications such as (i) mapping broad richness and community-composition patterns, (ii) identifying large-scale hotspots and coldspots under different climate scenarios, and (iii) informing high-level prioritization or reporting (e.g. national assessments, EU-wide strategies). We explicitly acknowledge in our conclusion and discussion Section 5 that local habitat management and single-site reserve design will require downscaling or complementary, higher-resolution models, and that BioAnalyst should be seen as a macroecological simulator rather than a replacement for fine-grain SDMs.
>
> König, Christian, et al. "Scale dependency of joint species distribution models challenges interpretation of biotic interactions." Journal of Biogeography 48.7 (2021): 1541-1551https://doi.org/10.1111/jbi.14106

---

> ### Author Response · Authors · 2025-11-19
> **Reply to questions of Reviewer 2dxb Part 1/2**
>
> We would like to thank the reviewer for the clarification questions which we answer one by one below.
> ### Question 1: What is the dimensionality of each data modality? How are the different modalities built?
> he dimensionality of each data modality is now available on Appendix B.2 on Table 2.
> Each modality is not build by us, rather than sampled from BioCube. We build Batches – a way to spatio-temporally organize and align the multi-modal data into an easier-to-handle format that accounts for 2 timesteps. Each Batch is a tuple dictionary containing all modalities (variable groups) with their respective variables for two consecutive months (time-steps). The Batch creation workflow is available on Appendix B.3.
> Coming to your question on Weakness 2 “For instance, Table 2 displays a list of species and their total number of occurrences. Does this mean that only occurrences of these species are used as input? Are these counts simply aggregated over 0.25 deg grid?”: The species variable group contains 28 species, and the number on the table is the total number of occurances for each species on all 233 monthly samples spanning from 01-01-2020 to 01-06-2020. More specific, the total number of occurances is an integer value that represents the species occurance as recorded and obtained from GBIF (according to BioCube), mapped on the coordinate grid we use with dimensions [280,160]. This variable group is one of the other 11 variable groups used as input to BioAnalyst and the complete list can be found on B.2, Table 2.
> For the next part of your very accurate observation: “. Do all variables have 10 levels? (not just pressure?). If there are two nested sums, this means all variables and level are summed together? I have spent some time on this (again, both on this paper and the original BioCube), and I still don’t see how exactly the input data looks like.” .
> Not all variables have 10 levels. Only the atmoshperic variable group contains pressure levels which are 13 in total not 10. In addition we would like to thank you for noticing our equation error when trying to represent the dataset samples, and we have reworked the section “Pre-training data selection”.Now it reads:
> … Selecting a Data Sample from the Data Batch yields a composite multi-modal cell of European coordinates, with a specific monthly time-stamp. Each of these Data Points for $t\in [0,233]$ contains 124 observations per location cell. The total observations are calculated by summing the 11 variable groups with the number of variables they contain and for the atmospheric variables multiplying the variables with the number of pressure levels and adding them to the sum.
> More specifically, we denote the observed data points at a discrete time $t$ by a collection of tensors $\mathbf{X}_t$:
>
> \begin{equation}
> \mathbf{X}_t \in \mathbb{R}^{\mathcal{H} \times \mathcal{W} \times F},
> \end{equation}
>
> with $\mathcal{H} = 160$, $\mathcal{W} = 280$, and $F = 124$ feature channels per spatial cell. We consider $I = 11$ variable groups indexed by $i = 1,\dots,I$, where the $i$-th group at time $t$ is represented as
> \begin{equation}
> V_t^{(i)} \in \mathbb{R}^{\mathcal{H} \times \mathcal{W} \times L_i},
> \end{equation}
>
> with $L_i$ the number of levels associated with that group (e.g.\ 13 pressure levels for atmospheric variables or a single level for all other variable groups). A full Data Sample is obtained by concatenating all variable-group tensors along the feature dimension,
> $X_t = concat_{i=1}^{I} V_t^{(i)}$
>
>
> so that each spatial cell $(\mathcal{H},\mathcal{W})$ at time $t$ is associated with a feature vector $x_{t,h,w}^{v} \in \mathbb{R}^{124}, v \in V$.
>
> During pre-training and fine-tuning, the input data consist of tuples of two consecutive time-steps. A generic input at time index $t$ is given by
> $(X_t, X_{t+1}) \\in (R^{H \\times W \\times F})^2$
>
>
> We hope that this makes our data sampling, Batch creation and model input more clear.

---

> ### Author Response · Authors · 2025-11-19
> **Reply to questions of Reviewer 2dxb Part 2/2**
>
> ### Question 2: How are the PerceiverIO and Swin Transformer architectures put together? (given that the first provides non-spatial representations and the second requires explicitly spatial ones). Other than this, I believe there is still a lot to do wrt weakness 1 in order to sustain the claim that the proposed model is a foundation model.
> As answering this question is also related to weakness 3, we have provided our answer above. Regarding weakness 1 connection, we have an answer in-place above too.
> We appreciate the reviewer’s concern and agree that the label “foundation model” should be used carefully. We have therefore clarified in the manuscript what we mean by this term and how BioAnalyst fits within current definitions.
> Following Bommasani et al. (2021) and subsequent work on environmental and ecological foundation models, we use “foundation model” to denote a model that is (i) trained on broad data at scale (often multimodal) using a generic objective, and (ii) designed so that its learned representations can be efficiently adapted to a range of downstream tasks within a domain. In this sense, our claim is intentionally domain-specific: BioAnalyst is a foundation model for European biodiversity rather than a general-purpose AI system.
> Concretely, BioAnalyst is pre-trained on a large multimodal ecological dataset that aligns long-term climate (ERA5), environmental and land-use variables, vegetation indices and species occurrence records and other modality groups over Europe at 0.25° resolution. The model learns generic spatio-temporal embeddings of ecological state, and we show that these embeddings can be adapted with lightweight heads to multiple qualitatively different tasks, including joint species distribution modelling for hundreds of plant species and abiotic climate prediction via linear probing. This pattern of broad multimodal pre-training plus reusable representations for diverse downstream tasks is precisely what characterises Earth-system foundation models such as Aurora in the climate community.
> At the same time, we fully acknowledge that BioAnalyst is an early biodiversity foundation model: our current evaluation covers only a limited set of downstream tasks, focuses on European plant communities and does not yet explore the full range of potential applications (e.g. invasive species detection, global transfer). We have revised the text to make this scope explicit and to describe BioAnalyst as a domain-specific foundation model for regional to national-scale biodiversity analysis, rather than implying that it already fulfils all possible criteria of a mature, general-purpose foundation model. We hope that this clarification addresses the reviewer’s concern while retaining the conceptual framing that motivates our design and experimental setup.

---

> ### Comment · Reviewer_2dxb · 2025-11-21
>
> I thank the authors for their response to W1. I agree with them that climate captures important abiotic factors conditioning biodiversity. However, I still think that many more downstream tasks are needed in order to claim that a model is a foundation model. The authors themselves seem to agree with this weakness by writing that a foundation model "can be efficiently adapted to a range of downstream tasks within a domain". They also write: "we fully acknowledge that BioAnalyst is an early biodiversity foundation model", which I interpret as an acknowledgement that this is preliminary work.
> In addition, the authors don't delve into why Aurora is not applied to the climate recovery task, where it should be competitive.
>
> Similarly, I find the issue I raised in W3 quite serious. Rearranging the latent tokens of PerceiverIO and treating them as spatial tokens is not well justified. SwinTransformers, as CNNs, assume a special relationship between neighbouring tokens. Why would this apply to the latent tokens of PerceiverIO? These tokens are arranged in a fully arbitrary manner. The authors claim that SwinT is suitable due to the multi-scale nature of many of the potential downstream tasks, but I don't see how this justification applies when the initial tensor given to SwinT has not particular spatial meaning. I am under the impression that this PerceiverIO+SwinT combination is a poor choice. I could, of course, be missing something (and thus be wrong about this), but the authors don't provide any convincing argument in this direction.

---

> > ### Author Response · Authors · 2025-11-22
> > **Answer to follow-up question of Reviewer 2dxb Part 2/2**
> >
> > We would like to thank Reviewer 2dxb for coming back to this important part of our architecture. Below our arguments.
> >
> > We respectfully do not agree with the premise that latent tokens are “arranged in a fully arbitrary”. The latent representations possess explicit structure through three mechanisms that we consider as making the Swin processing choice as well-justified.
> >
> > ### Structured latent organization
> > The encoder initializes latent parameters with explicit spatial organization: each modality group $m \in \{1, \ldots, M \}$ receives $N_p$ latent tokens corresponding to the $N_p$ spatial patches in the input grid. These are concatenated in a deterministic order: These are concatenated in a deterministic order:
> >
> > $z = [z_1^{(1)}, ..., z_{N_p}^{(1)}, z_1^{(2)}, ..., z_{N_p}^{(2)}, ..., z_{N_p}^{(M)}] \\in R^{M \\cdot N_p \\times D}$
> >
> > where $\mathbf{z}_i^{(m)}$ is the latent token for modality $m$ at spatial location $i$. In our implementation, with $M=23$ modality groups and $N_p = 700$ patches (corresponding to a $20 \times 35$ spatial grid), the total sequence length is $23 \times 700 = 16100$. When reshaped to $(M, H_p, W_p)$ = (23, 20, 35), each depth slice corresponds to one modality group, and within each slice, the $(H_p, W_p)$ dimensions preserve the original spatial patch topology.
> >
> > ### Position encodings enforce spatial relationships
> > Given standard practices in vision transformers, we also inject explicit spatial information via Fourier position encodigns. For each patch at spatial location $(h, w)$ with geographic coordinates $(\phi_{h, w}, \lambda_{h,w})$ (latitude, longitude), we compute:
> > $p_{h,w} = \text{FourierEnc}(\phi_{h,w}, \lambda_{h,w}) \in \mathbb{R}^D$
> >
> > These position encodings are added to the token embeddings before Perceiver IO processing: $\tilde z_{i}^{(m)} = z_i^{(m)} + p_{h(i), w(i)}$
> >
> > where $h(i), w(i)$ maps the linear patch index $i$ to spatial coordinates. Importantly, neighboring tokens in the reshaped grid $(h, w)$ and $(h’ , w’)$ receive position encoding corresponding to geographically adjacent patches.  The Fourier encoding ensures that $|| p_{h,w} - p_{h', w'} ||$
> >   is small when patches are spatially close, providing the spatial continuity that Swin’s window-base attention exploits.
> >
> > ### Spatial coherence from inductive bias
> > The swin backbone imposes a spatial inductive bias through window-based attention: tokens within local windows $(c, h, w) \in [c_0, c_0+W_c] \times [h_0, h_0+W_h] \times [w_0, w_0+W_w]$ interact via self-attention, assuming spatial locality. During training, the model learns latent representations that align with this bias. The position encodings provide the coordinate system, and the gradient signals from Swin’s spatial operators encourage the Perceiver IO encoder to organize information spatially coherently.
> >
> > We also attempted to validate this spatial structure through attention analysis: visualization of encoder cross-attention patterns revealed geographically organized focus regions (e.g., species attention concentrates on Scandinavia and Central/Western Europe, similarly to our actual data points; appendix D, Figure 12), not random distributions. This could confirm that latents encode meaningful spatial relationships compatible with Swin’s architectural assumptions.
> >
> > ### Why multi-scale processing applies
> > The reviewer questions whether multi-scale justification applies “when the initial tensor has no particular spatial meaning”. However, we would like to state that the tensor does have spatial meaning:
> > the $(H_p, W_p)$ dimensions represent the geographic grid
> > position encodeings encode absolute coordinates
> > the depth dimensions $M$ organizes modalities, which exhibit natural hierarchies (e.g., atmospheric levels at different pressure scales)
> > Swin’s hierarchical downsampling operates on the spatial dimensions, progressively aggregating information from finer spatial scales (individual patches) to coarse scales (regional patterns), which directly corresponds to the multi-scale nature of ecological processes where local habitat conditions and regional climate jointly determine biodiversity.
> >
> > We hope that this clarifies our design choices and answers your questions about the architecture.
> >
> >
> > References for Part 1/2:
> >
> > [1] https://digital-strategy.ec.europa.eu/en/faqs/general-purpose-ai-models-ai-act-questions-answers
> >
> > [2] https://hai.stanford.edu/news/what-foundation-model-explainer-non-experts
> >
> > [3] Bommasani, R., Hau, A., Klyman, K., & Liang, P. Foundation Models and the EU AI Act. In NeurIPS 2024 Workshop on Regulatable ML.
> >
> > [4] Bommasani, R. (2021). On the opportunities and risks of foundation models. arXiv preprint arXiv:2108.07258.

---

> ### Author Response · Authors · 2025-11-22
> **Answer to follow-up question of Reviewer 2dxb Part 1/2**
>
> Dear Reviewer 2dxb. Thank you for the follow-up questions and we appreciate the opportunity to clarify what we mean by “foundation model” and how BioAnalyst fits that definition.
>
> In line with the definition introduced by the Stanford Center for Research on Foundation Models and widely adopted since, we use “foundation model” to refer to a model trained on broad data at scale that can be adapted (e.g. fine-tuned) to a wide range of downstream tasks, rather than to a model for which all possible downstream tasks have already been exhaustively demonstrated. This understanding is consistent with both AI theory work and policy definitions (e.g. the EU AI Act), which emphasise the training regime and adaptability on broad data, not a particular number of benchmark tasks [1], [2], [3], [4].
>
>
> In that sense, BioAnalyst follows the same pattern as other domain-specific foundation models in geosciences. For example, Prithvi WxC is described as “a foundation model for weather and climate”, pretrained on heterogeneous climate data and then fine-tuned on a finite set of weather and climate tasks (forecasting at different lead times, downscaling, projections). Similarly, Aurora is presented as “a foundation model for the Earth system” trained on more than one million hours of geophysical data and then specialised for a limited but diverse set of downstream applications (high-resolution weather, waves, air quality, cyclone tracks). In both cases, the models are called foundation models because they are broadly pretrained and can be adapted to many tasks, not because all such tasks have already been instantiated in the initial paper.
> BioAnalyst fits this same pattern within the biodiversity domain. It is pretrained on long-term, multimodal ecological data (species occurrences, climate, land use, vegetation, etc.) over Europe, using a generic spatio-temporal objective to learn reusable embeddings of ecological state. In the present work we demonstrate adaptation of these embeddings to several qualitatively different downstream tasks: joint species distribution modelling for different species category than the pre-training (plants vs animals), community-level richness prediction, and abiotic climate prediction via linear probing. This already illustrates that the learned representations support distinct biotic and abiotic tasks without retraining the model from scratch, which is precisely the behaviour we expect from a domain-specific foundation model.
> We fully agree that there are many more downstream tasks that could be explored (e.g. invasive species risk, extinction risk, ecosystem state indicators, cross-region transfer), and we see these as natural directions for future work rather than prerequisites for using the “foundation model” terminology.
> The wording “early biodiversity foundation model” appeared only in our previous response to Question 2 , not in the manuscript itself. Our intention there was to convey that BioAnalyst belongs to a new and rapidly emerging class of biodiversity-focused foundation models, not to suggest that it “is not yet” a foundation model in the technical sense or that the work is merely preliminary.
>
> Regarding your next remark *the authors don't delve into why Aurora is not applied to the climate recovery task, where it should be competitive.*
> That is not accurate. We are comparing BioAnalyst with Aurora and have introduced extra baselines after reviewers comments for all downstream tasks. More specific, the results for all downstream tasks are available at Table 1, line 456 in PDF file. The climate linear probing task was done with temperature and precipitation. The reason Aurora is not competitive here is probably because it was not pretrained with precipitation as an input variable (see https://microforest.github.io/aurora/batch.html). In contrast, our linear baselines and models use both temperature and precipitation, making comparisons more aligned with the climate recovery task.
> Also, in our text we explicitly mention the comparison: “In the same theme, comparing BioAnalyst with baselines such as the pre-trained Aurora-025, a Random Forest model and Support Vector Machine yielded a stronger predictive capacity for BioAnalyst. However, this comparison should be interpreted only as an indicative baseline: BioAnalyst is pre-trained directly on monthly biodiversity states, whereas Aurora is an Earth-system foundation model that is pre-trained to forecast geophysical variables at short lead times (typically 6-hour steps) and fine-tuned for weather, wave and air-quality tasks rather than biodiversity.”
>
> We hope these clarifications resolves the concern.
>
> References are added on the Part 2/2 due to limited space.

---

> ### Author Response · Authors · 2025-11-26
> **Rebuttal period is concluding soon**
>
> Dear reviewer 2dxb,
>
> Thank you again for the thoughtful feedback and constructive comments. As the rebuttal deadline is approaching, we would be grateful if you could take a look at our rebuttal and the updated text to ensure we have addressed your concerns accurately. If you have additional remarks, questions or requests, we are more than happy to address them! We appreciate your time and consideration.

---

### Official Review · Reviewer_GJx9 · 2025-10-29

**Soundness:** 2
**Presentation:** 2
**Contribution:** 3
**Rating:** 2
**Confidence:** 3

**Summary:**

This paper introduces BioAnalyst, a multimodal foundation model for biodiversity analysis and ecological forecasting.
The model integrates ten heterogeneous data modalities — including climate, vegetation, soil, and species occurrence variables — through a hybrid architecture combining a Perceiver IO encoder–decoder with a 3D Swin Transformer backbone.
BioAnalyst is pretrained on a 20-year subset (2000–2020) of the BioCube dataset at 0.25° spatial resolution over Europe and is fine-tuned for three downstream tasks: (1) biodiversity rollout forecasting, (2) species distribution modeling, and (3) climate variable reconstruction (linear probing).
The authors claim that BioAnalyst outperforms Microsoft’s Aurora climate foundation model, especially in data-scarce scenarios, and publicly release model weights and code.

**Strengths:**

- Timely and societally valuable problem. Bringing foundation-model methodology to biodiversity and conservation is meaningful and impactful for environmental science.

- Ambitious multimodal integration. The model aligns diverse ecological variables within a single latent representation, a nontrivial data-engineering achievement.

- Transparent and reproducible. Code, model weights, and detailed appendices are provided.

- Clear writing and motivation. The introduction and related-work sections convincingly position biodiversity modeling as an under-served domain for AI FMs.

**Weaknesses:**

- Engineering integration rather than scientific novelty.
The architecture is an adaptation of existing components (Perceiver IO, Swin Transformer) with minor modifications and a temporal-difference loss.
While this is solid engineering, it lacks new methodological ideas or theoretical insights expected at ICLR.

- Over-reliance on outperforming Aurora.
The only strong empirical claim is that BioAnalyst “outperforms” Aurora on a few metrics.
However, Aurora is primarily a climate model, not a biodiversity model, and its temporal/spatial resolutions differ.
Therefore, this comparison is not fully fair, and exceeding it on select metrics does not establish a significant scientific contribution.
In short, beating Aurora is not, by itself, enough to justify an ICLR paper.

- Inconsistent metric reporting.
The authors report 𝑅2 for the abiotic task (where results are strong) but omit it for the biotic fine-tuning task, where performance is less clear.
This selective reporting weakens confidence in the claim of superiority.

- Limited empirical rigor.
No ablations on architecture or modalities, and no additional baselines beyond Aurora.
The F1 difference (0.996 vs 0.994) is negligible, and uncertainty quantification is missing.

- Narrow evaluation scope.
The dataset covers only terrestrial Europe at 0.25° resolution, and the model is not tested globally or across unseen taxa.
This makes it difficult to assess generalization and robustness.

- Scientific framing.
The paper presents a well-engineered pipeline but does not reveal new scientific understanding or modeling principle.
The contribution lies in integration and system building rather than discovery.

**Questions:**

- Report consistent metrics (including 𝑅2, correlation, and uncertainty) for both biotic and abiotic tasks.

- Include additional baselines and ablation studies, e.g., single-modality or reduced-modality versions.

- Clarify the ecological interpretation of what BioAnalyst learns (e.g., which cross-modal relationships drive predictions).

- Temper the Aurora comparison, acknowledging task mismatches and limited evidence.

- Consider submission to a domain-specific venue such as Environmental Modelling & Software, Ecological Informatics, or Nature Scientific Data, where the work’s environmental and integrative value would be more impactful and appropriately appreciated.

---

> ### Author Response · Authors · 2025-11-19
> **Reply to weaknesses of Reviewer GJx9 Part 1/2**
>
> We would like to thank the reviewer for their time spend to read our manuscript and provide us with accurate remarks that improved our manuscript. We will cover all the Weaknesses one by one in this part.
> ### Weakness 1: Engineering integration rather than scientific novelty. The architecture is an adaptation of existing components (Perceiver IO, Swin Transformer) with minor modifications and a temporal-difference loss. While this is solid engineering, it lacks new methodological ideas or theoretical insights expected at ICLR.
> We acknowledge the reviewers remark. We have submitted our manuscript to the “applications to physical sciences (physics, chemistry, biology, etc.)” track for the reason that we demonstrate the application for this composite architecture to a completely new domain and data modalities. More specific we believe that our contributions are of importance and relevant to ICLR:
> - Development of the first Multi-modal Biodiversity Foundation Model: To our knowledge the first large-scale AI model tailored for biodiversity modelling, capable of processing and integrating diverse data types to model complex ecological phenomena.
> - Advancement in Predictive Biodiversity Analytics: We demonstrate BioAnalyst's predictive capabilities in key applications such as species distribution modelling, biotic and abiotic reconstruction, and population trend forecasting, especially in data-scarce scenarios.
> - Open Collaboration and Resource Sharing: By openly releasing BioAnalyst's code, weights, and fine-tuning workflows, we aim to foster collaborative efforts within the scientific community, thereby accelerating research and conservation initiatives that address pressing ecological challenges.
>
> ### Weakness 2: Over-reliance on outperforming Aurora. The only strong empirical claim is that BioAnalyst “outperforms” Aurora on a few metrics. However, Aurora is primarily a climate model, not a biodiversity model, and its temporal/spatial resolutions differ. Therefore, this comparison is not fully fair, and exceeding it on select metrics does not establish a significant scientific contribution. In short, beating Aurora is not, by itself, enough to justify an ICLR paper.
>
> We put emphasis on the Aurora because a) it uses the same architecture as BioAnalyst (Percieve IO for Encoder-Decoder  & Swin Transformer for Backbone) for pretraining, b) Aurora was also pretrained using ERA5 data, like BioAnalyst, so some modalities are similar, and c) climate is the fundamentally detriment for ecological niches. We have addressed point c extensively in our reply on Reviewer 2dxb Weakness 1, and you can also see this effect in Figure 9 and Figure 10 where the correlation between the cross-attention of climate and species is very high.
> We have made explicit in our revised text the differences between Aurora and BioAnalyst, so the reader can spot them immediately and avoid wrong conclusions.
> ### Weakness 3: Inconsistent metric reporting. The authors report 𝑅2 for the abiotic task (where results are strong) but omit it for the biotic fine-tuning task, where performance is less clear. This selective reporting weakens confidence in the claim of superiority.
> We believe there is a confusion here. We report R2 for the abiotic (climate-probing) task because it is a continuous regression problem, which makes R2 a standard and interpretable measure of explained variance. By contrast, our biotic fine-tuning task involves joint species modelling and temporal forecasting of presence/absence or occurrence probabilities across multiple taxa- an inherently classification or multi-label forecasting task, where R2 is not conventionally used (and may even be misleading) because the target is discrete rather than continuous. Thus, we report appropriate classification metrics (e.g., GeolifeCLEF F1, RMSE of probabilities) for the biotic task rather R2 . We have clarified this important distinction in the revised manuscript, explicitly noting the difference in target types and metric appropriateness between the two tasks to avoid any perceived inconsistency, both in main text and on appendix.

---

> ### Author Response · Authors · 2025-11-19
> **Reply to weaknesses of Reviewer GJx9 Part 2/2**
>
> We continue the replies to the Weaknesses.
> ### Weakness 4: Limited empirical rigor. No ablations on architecture or modalities, and no additional baselines beyond Aurora. The F1 difference (0.996 vs 0.994) is negligible, and uncertainty quantification is missing.
> Thank you for that critical remark that triggered us to further enhance our manuscript.
> Based on your remarks, we have performed the below:
> a) We have tested a different backbone MViT and its implementation is available in our codebase, although its performance was very poor from initial experiments and we omit to include it in our results.
> b) We have tested 4 different combinations of modalities and report their results abstractly on Subsection 4.1 and in more detail on Section D. - BioAnalyst ablations and interpretability. Table 9 highlights this comparison for the species distribution task with the pre-trained BioAnalyst.
> 1) [Species, Surface]
> 2) [Species, Climate]
> 3) [Species, RLI]
> 4) [Species, Edaphic]
> 5) [Species, Edaphic, RLI, Climate, Surface]
>  c) We have added 2 more baselines, one for each downstream task, and we have updated the result section of our manuscript to account for these new results.
> Regarding the uncertainty quantification, that was not part of our original work plan, although we recognise its importance, and we explicitly state that on our Discussion part as a future work element.
> As climate is crucial and no similar models trained with the exact same modalities as BioAnaylst exist, we thought only a climate Foundation Model (i.e. Aurora) is a valid comparison with BioAnalyst as it captures the variation in ecological niches, especially since Aurora and BFM both used ERA-5 data for climate modality during pretraining. AlphaEarth (and other GeoFMs) was trained on remote sensing data based on multispectral optical instruments. A direct comparison cannot be made as the model represents very different physical processes.
>
> ### Weakness 5: Narrow evaluation scope. The dataset covers only terrestrial Europe at 0.25° resolution, and the model is not tested globally or across unseen taxa. This makes it difficult to assess generalization and robustness.
> BioAnalyst is pre-trained for European only coordinates. The model has been further evaluated by adding additional baselines to compare with (see previous weakness).
>
> ### Weakness 6: Scientific framing. The paper presents a well-engineered pipeline but does not reveal new scientific understanding or modeling principle. The contribution lies in integration and system building rather than discovery.
> Big part is also defended as a reply on the first weakness.
> Also, BioAnalyst builds a representational layer via modalities other FMs have not addressed as far as we know. This representational layer opens new avenues for deep learning in ecological modelling, that purely remote sensing based foundation models cannot address. We never claimed to introduce a new method, but we qualify that BioAnalyst opens avenues for challenging ecological modelling tasks. In addition, we strongly believe that our development and sound reporting can be the springboard of a new line of scientific discoveries on AI-driven ecology that can further improve our world.

---

> ### Author Response · Authors · 2025-11-19
> **Reply to questions of Reviewer GJx9 Part 1/2**
>
> We would like to thank again the Reviewer GJx9 for their constructive questions. Below we reply in-place on all the questions.
>
> ### Question 1: Report consistent metrics (including 𝑅2, correlation, and uncertainty) for both biotic and abiotic tasks.
> See our explanation above (Weakness 3), there is (in fact) no issue of consistency here. The tasks are inherently different from a machine learning perspective (regression vs classification).
>
> ### Question 2: Include additional baselines and ablation studies, e.g., single-modality or reduced-modality versions.
>  We have drafted a new section D. BioAnalyst Ablations and Interpretability where we highlight the performance of 5 combinations of modalities on Table 9. We have also drafted a new subsection C.4.4. - Baselines, where we list in detail the baselines used for our experiments, their limitations and the reasoning behind their selection. We have updated our main text to account for them.
>
> ### Question 3: Clarify the ecological interpretation of what BioAnalyst learns (e.g., which cross-modal relationships drive predictions).
>
> We would like to thank the reviewer for this critical remark which is also similar to the reviewer gCqn question. We repeat our reply here and enhance the last part to further account for your question: We have made updates on our manuscript to account for the new insights from investigating BioAnalysts inner modalities. More specifically, we have drafted a new section on Appendix D. - BioAnalysts Interpretability:  where we discuss our findings from looking into the cross-attention attention maps for all modality groups. We have introduced 3 new Figures, more specific:
> Figure 9: Mean attention weight per modality group contributions for 12 test windows (months).
> Figure 10: Mean attention weight per modality group contribution for the Encoders cross-attention layer.
> Figure 11: Cross-modality attention correlation matrix for all the modality groups with their respective variables.
> Figure 12: Cross-modality spatial attention map for the species variable group.
> Coming to the question in more detail, from Figure 10: we see that the most contributing groups for the biodiversity modelling task are: climate, species, surface, atmospheric level 1 & 12.
> We can better clarify what BioAnalyst learns by looking at Figure 10 in more detail and the cross-modality correlation matrix. There, we see that vegetation, land, agriculture, forest, redlist, misc and some atmospheric levels from the modality groups correlate strongly with each other. Species variables seem to correlate mildly with surface edaphic.
> From our ablation experiment we see that: while the more comprehensive models (C5 and BFM Full) provide consistent predictions across physical variables (Surface, Edaphic, and Climate groups), single-modality combinations can offer slightly better performance in specific tasks. For example, for species abundance predictions (bottom section), the Climate-only model (\textbf{C1}) frequently outperforms the fuller models (e.g., for species ID 9809229, C1 achieves an MAE of 0.627 compared to 0.740 for C5). This could suggest that either climatic variables are the dominant helpers in predicting species distributions in this dataset, and adding additional modalities (as in C5) may occasionally introduce noise rather than distinct predictive signals for certain species.
>
> ### Question 4: Answered on the next part

---

> ### Author Response · Authors · 2025-11-19
> **Reply to questions of Reviewer GJx9 Part 2/2**
>
> ### Question 4: Temper the Aurora comparison, acknowledging task mismatches and limited evidence.
> We appreciate this comment and agree that the comparison to Aurora should be interpreted with caution. Aurora is an Earth-system foundation model trained on over a million hours of heterogeneous geophysical data and primarily optimised for short-lead (6-hour) forecasts of atmospheric and oceanic variables, which is then fine-tuned for operational weather, wave and air-quality forecasting tasks. The official documentation also emphasises that the fine-tuned Aurora variants are intended for specific datasets (e.g. IFS HRES or CAMS), and that performance may degrade when applied outside those domains.
>  In contrast, BioAnalyst is pretrained directly on monthly biodiversity targets and multimodal ecological covariates (land use, vegetation, species records, etc.), and is explicitly designed for regional biodiversity prediction. Our use of Aurora-0.25° therefore constitutes an adaptation of a climate/atmosphere foundation model to a downstream task it was not specifically designed or fine-tuned for.
> We have revised the manuscript to make this clear. In the Results section we now state that the Aurora baseline is indicative rather than fully comparable, and that our evidence is limited to a small set of biodiversity tasks in Europe. We explicitly avoid general claims about Aurora’s overall capability and instead frame the comparison as showing that, under a straightforward adaptation, a biodiversity-specialised, multimodal foundation model (BioAnalyst) currently achieves better performance on our specific ecological prediction tasks than a generic Earth-system foundation model. We see these approaches as complementary rather than competing.
> We have further drafted a new subsection C.4.4. - Baselines, where we detail our reasoning for the selection and the abovementioned mismatches.
>
> ### Question 5: Consider submission to a domain-specific venue such as Environmental Modelling & Software, Ecological Informatics, or Nature Scientific Data, where the work’s environmental and integrative value would be more impactful and appropriately appreciated.
> We thank you for this suggestion! Still, we believe that our contribution to ICLR is of significance, that is why we have spend a great amount of time to improve and further enhance our manuscript, taking into consideration all the remarks, weaknesses and questions of all reviewers.

---

> ### Author Response · Authors · 2025-11-26
> **Rebuttal period is concluding soon**
>
> Dear reviewer GJx9,
>
> Thank you again for the thoughtful feedback and constructive comments. As the rebuttal deadline is approaching, we would be grateful if you could take a look at our rebuttal and the updated text to ensure we have addressed your concerns accurately. If you have additional remarks, questions or requests, we are more than happy to address them! We appreciate your time and consideration.

---

### Official Review · Reviewer_gCqn · 2025-11-01

**Soundness:** 3
**Presentation:** 3
**Contribution:** 3
**Rating:** 6
**Confidence:** 3

**Summary:**

The paper presents bioanalyst, a transformer-based, multimodal foundation model for biodiversity. It integrates diverse geospatial and ecological data and reports strong results on forecasting tasks.

**Strengths:**

- integrates diverse data types (remote sensing, climate, soils, species, etc.).

- code is available and opensourced

- comparison with microsoft’s aurora gives an initial external reference point.

- generally well-written with detailed implementation details.

**Weaknesses:**

- Only two downstream tasks are shown (species forecasting; climate probing), with limited baselines (Aurora only).

- Can include task-specific baselines.

- why a swin transformer? please justify the backbone choice.

- How are the encoders-decoders trained? Together with the whole model or are they pretrained beforehand?

- There is no uncertainty quantifcation discussed in the work. It would be good to have some experiments around that.

- Has the model been trained to predict x_{t+1} directly (next-step forecasting)? how sensitive are results to this choice? What if it is trained with x_{t+2} or if it is varied during training?

- The paper would benefit with some analysis on which features impact the model more? Perhaps a qualitative analysis.
-  The work would further benefit from more out of distribution experiments and zershot analysis being a foundation model

**Questions:**

- What is the contribution of each modality group (climate/soil/land/NDVI/species) to downstream performance?

- why a swin transformer over alternatives for spatiotemporal data?

- can you show zero-shot performance on out-of-distribution regions/years and compare against aurora

- do the features/modalities share the same grid/temporal resolution? if not, what resampling/normalization/alignment steps are used before fusion?

- which datasets were used for fine-tuning each downstream task? is there any species/time overlap with pretraining that could bias results?

- Please clarify the rollout loss.

---

> ### Author Response · Authors · 2025-11-19
> **Reply to weaknesses of Reviewer gCqn**
>
> We would like to thank the reviewer for their constructive criticism and accurate remarks. We will cover all the Weaknesses and Questions one by one.
> ### Weakness 1&2: Only two downstream tasks are shown (species forecasting; climate probing), with limited baselines (Aurora only). Can include task-specific baselines.
> We agree that featuring only two downstream tasks (species forecasting and climate probing) and comparing primarily to the Aurora baseline does limit the breadth of our evaluation. However, we deliberately chose these tasks to showcase the dual applicability of BioAnalyst – both biotic (species distributions) and abiotic (climate structure) – because they exemplify two very different modelling regimes (multi‑label classification vs. continuous regression), and there remain very few established foundation‑model baselines in ecology. It is also worth emphasising that our biotic-task is more than a traditional single‐species distribution model: we fine-tune the model for joint species modelling across multiple taxa, and simultaneously perform temporal forecasting of species distributions over successive time steps. In other words, the task intrinsically captures variation not only in species-environment relationships but also in community composition and its temporal dynamics, thereby exercising the model’s ability to represent multi-species interactions, temporal trends and spatial-ecological structure in one unified experiment.
> For baselines, for the climate linear probing task, we have added both Random Forest and SVM models with their results already updated in our manuscript updated version. For the Joint Species Distribution Modelling task, work is in progress and the results will become available hopefully on the last version of the manuscript.
> ### Weakness 3: why a swin transformer? please justify the backbone choice.
> The Swin Transformer was chosen because biodiversity patterns can be observed at multiple spatial scales (e.g., from microclimates to continental zones), and Swin’s hierarchical architecture with patch merging/splitting appeared as an interesting and suitable choice for a backbone that would ingest data with multi-scale structuring. Additionally, its window-based attention mechanism also is stated as having linear complexity (when compared to the original ViT) in processing image/image-like inputs, making it a computationally efficient choice - a property that certainly would be desirable in the case of processing/training/testing on large inputs (e.g., input data spanning our entire planet, not just continental Europe).  Additionally, we adjusted the architecture to support temporal conditioning through lead-time embeddings that can modulate processing at each stage, allowing the model to adapt representations based on the prediction time horizon, which is a capability we use for multi-month forecasting.
> We have updated our Appendix, and more specifically we drafted a new subsection A.2 - Design Choices that clarifies our choice.
> ### Weakness 4: How are the encoders-decoders trained? Together with the whole model or are they pretrained beforehand?
> The encoders-decoders are trained together along with the backbone. We have updated our text on C.3 to account for that. Now it reads: “We train the complete architecture together, both encoder-decoder and backbone.”
> ### Weakness 5: There is no uncertainty quantifcation discussed in the work. It would be good to have some experiments around that.
> Regarding the uncertainty quantification, that was not part of our original work plan, although we recognise its importance, and we explicitly state that on our Discussion part as a future work element.
> ### Weakness 6: Has the model been trained to predict x_{t+1} directly (next-step forecasting)? how sensitive are results to this choice? What if it is trained with x_{t+2} or if it is varied during training?
> BioAnalyst has been pre-trained to predict $x_{t+1}$ by providing as input $[x_{t-1}, x_{t}]$. During fine-tuning and more specific, roll-out fine-tuning, BioAnalyst was extended to predict 6 and 12 steps ahead in 2 stages. At the first stage, we fine-tune the complete BioAnalyst, adding VeRA adapters on the backbone for 12000 steps with a horizon of 6 time-steps (6 months), using constant learning rate of $0.00003$, weight decay of $0.000003$ and store the weights. In the second stage we load the previously stored weights and continue the roll-out fine-tuning for 12000 time-steps more, with a horizon of 12 steps (1 year), using constant learning rate of $0.00001$ and weight decay of $0.000001$. Fine-tuning with a mix of variable horizons is a technique used on Aurora \citep{bodnar2024aurora}, PrithviWxC \citep{schmude2024prithviwxcfoundationmodel} and Aardvak \citep{allen2025end}, offering improved long-horizon forecasting capabilities, which we also observed ourselves and is highlighted on Figure 3 for the species variables.
> ### Weakness 7&8: Related with Questions, answers below

---

> ### Author Response · Authors · 2025-11-19
> **Reply to questions of Reviewer gCqn Part 1/2**
>
> Again, we would like to thank the reviewer for their critical questions, that allowed us to improve and clarify elements of our manuscript. We provide answers to the questions one by one.
> ### Question 1: What is the contribution of each modality group (climate/soil/land/NDVI/species) to downstream performance?
> We would like to thank the reviewer for this critical remark which is also related to the previously mentioned weakness as well as to a question from reviewer GJx9. In response, we have made updates on our manuscript to account for the new insights from investigating BioAnalysts inner modalities. More specific, we have drafted a new section on Appendix D. - BioAnalyst Ablations and Interpretability:  where we discuss our findings from looking into the cross-attention attention maps for all modality groups. We have introduced 4 new Figures and 1 Table, more specific:
> Figure 9: Mean attention weight per modality group contributions for 12 test windows (months).
> Figure 10: Mean attention weight per modality group contribution for the Encoders cross-attention layer.
> Figure 11: Cross-modality attention correlation matrix for all the modality groups with their respective variables.
> Figure 12: Cross-modality spatial attention map for the species variable group.
> Table 9: Performance comparison between different modality combinations
> Coming to the question in more detail, from Figure and 10: we see that the most contributing groups for the biodiversity modelling task are: climate, species, surface, atmospheric level 1 & 12.
> ### Question 2: why a swin transformer over alternatives for spatiotemporal data?
> This has been answered previously on Weakness 3.
> ### Question 3: can you show zero-shot performance on out-of-distribution regions/years and compare against aurora
> We are currently working on that, and will update our manuscript with the latest results when available.
> ### Question 4: do the features/modalities share the same grid/temporal resolution? if not, what resampling/normalization/alignment steps are used before fusion?
> Yes, all the variable groups with their respective variables share the same grid/temporal resolution. More specific, the modalities sampled from BioCube come with grid coordinate system WGS84, 0.25 degrees, and 1 month lead-time, starting on the 1st of each month. Before fusion we apply the following principles, also available on B.3. “Build Batches”:
> - Longitude coordinates are wrapped into the interval $(-180^{\circ}, 180^{\circ}]$ to ensure consistency across global datasets.
> - Timestamps are standardised to \texttt{datetime64} objects with monthly resolution.
> - Latitude and longitude coordinates are snapped to the $0.25^{\circ}$ grid to match the spatial resolution of the batch format.
> - Missing values are imputed with zeros, enabling compatibility with models that do not natively support NaN values.
> ### Question 5: which datasets were used for fine-tuning each downstream task? is there any species/time overlap with pretraining that could bias results?
> As mentioned in section 3.4 “More specific, the first task involves partial model adaptation: the BioAnalyst's encoder and decoder are frozen. At the same time, the backbone is fine-tuned with VeRA adapters using historical species plant occurrence (500 most commonly occurring species) data from the GeoLifeCLEF2024 benchmark dataset \citep{geolifeclef-2024} to forecast time-series distributions. The second task is diagnostic: a regression head is trained on top of frozen decoder embeddings for both BioAnalyst and Aurora, to predict monthly climate variables (2-meter temperature and precipitation) from CHELSA v2.1 \citep{karger2017climatologies}.”
>
> The CHELSA data was from the year 2000 till 2019.  The GeoLifeCLEF24 data is for years 2017-2020. Updated sections 3.4 and and B.4 to reflect this information.
> The BioAnalyst model modalities do cover the time period above (2000-2020), but none of the modalities (neither the species from GeoLifeCLEF or the CHELSA climatological data) are represented in the pre-training process. Hence, we do not see how there could be any inherent bias in the results of the downstream tasks.

---

> ### Author Response · Authors · 2025-11-19
> **Reply to questions of Reviewer gCqn Part 2/2**
>
> ### Question 6: Please clarify the rollout loss.
>
>
> We thank the reviewer for pointing out that our description of the rollout loss was unclear. We now explicitly define the multi-step autoregressive loss used during roll-out fine-tuning and correct the indexing in Eq. (10). Specifically, starting from an observed state , we unroll the model autoregressive for monthly steps and compute the average task-dependent loss over all predicted states:
>
>
> \textbf{Roll-out finetuning}: In this setting, we fine-tune the entire BioAnalyst for six and twelve rollout steps, effectively predicting biodiversity dynamics six months and one year ahead. Fine-tuning with a mix of variable horizons is a technique successfully used on Aurora \citep{bodnar2024aurora}, PrithviWxC \citep{schmude2024prithviwxcfoundationmodel} and Aardvak \citep{allen2025end}, enhancing long-horizon forecasting capabilities. In practice, we freeze the whole architecture, including the encoder, decoder, and backbone, while training only the newly added VeRA adapters \citep{kopiczko2024vera} on the backbone's attention heads. We found VeRA to perform equally or sometimes slightly better than other Parameter Efficient Fine-tuning Techniques (PEFTs), such as LoRA \citep{hu2022lora}, which use only one-tenth of the learnable parameters.
> More specific, starting from an observed state $x_t$, the model is unrolled autoregressively: at each step $k$ it takes the previous prediction $\hat x_{t+k-1}$ as input and produces a new prediction $\hat x_{t+k}$. We supervise all intermediate steps with the same task-dependent loss $L_{TD}$ and define the $K$-step rollout loss as
>
> $L_{rollout}(t,K) = \\frac{1}{K} \\sum_{k=1}^{K} L_{TD}( \hat x_{t+k}^{u}, x^{u}_{t+k}\)$
>
>
> Our mix horizons are $K=6$ and $K=12$ months. To keep training stable and affordable for long sequences, we follow the ``pushforward trick'' of \citet{brandstetter2022message}: during backpropagation we stop gradients at the inputs of steps $k>1$, so that the loss is averaged over all steps for evaluation, but only the final step ($k=K$) contributes gradients to the model parameters.

---

> ### Author Response · Authors · 2025-11-26
> **Rebuttal period is concluding soon**
>
> Dear reviewer gCqn,
> Thank you again for the thoughtful feedback and constructive comments. As the rebuttal deadline is approaching, we would be grateful if you could take a look at our rebuttal and the updated text to ensure we have addressed your concerns accurately. If you have additional remarks, questions or requests, we are more than happy to address them!
> We appreciate your time and consideration.

---

### Author Response · Authors · 2025-11-22
**Response to all Reviewers regarding additional baselines & ablation studies**

We would like to inform all the Reviewers that have compiled all their comments, responded in-place to all weaknesses and questions. We have performed a series of additional experiments, adding 4 new Baselines for the downstream task and a series of Ablations.
More specific:

We have performed additional experiments with 4 baselines, two for each downstream task as addition to Aurora. More specific, for the joint SDM task, we now report: (i) LatentMLP and (ii) ConvLSTM baseline results. For the climate linear probing task we report: (i) Random Forest and (ii) Support Vector Machine baseline results.
Table 1 (line 452 on PDF) has been updated to reflect this new baseline results and a new subsection on Appendix C.4.4 - Baselines (line 1686 in PDF) has been drafted to describe them.

In addition, we have performed a series of Ablation studies for BioAnalyst, (i) modality contributions (ii) variable group performance analysis. More specifically, we have produced 4 new Figures, highlighting the cross-attention weights, their correlations and their spatial coverage. For (ii) we have pre-trained BioAnalyst on 5 combinations of modalities (C1-C5) and report their performance on Table 9.

All these new results are under a newly drafted section D - BioAnalyst ablations and interpretability (line 1808 in PDF) and their contribution is embedded on main text in relevant sections.

**Table 1 –** Performance of the species distribution forecasting and linear probing tasks.
Note: $R^2$ is not applicable to the classification-style species forecasting task and therefore not reported.

| Model        | Species forecast Loss | Species forecast F1 | Species forecast RMSE | Linear probing Loss | Linear probing $R^2$ | Linear probing RMSE |
|-------------|-----------------------|---------------------|-----------------------|---------------------|----------------------|---------------------|
| BioAnalyst  | 0.0057                | 0.9964              | 0.5284                | 0.0225              | 0.9002               | 0.1499              |
| Aurora      | 0.0130                | 0.9945              | 0.5014                | 0.2668              | 0.7354               | 0.5144              |
| RandomForest| --                    | --                  | --                    | --                  | 0.7260               | 0.2426              |
| SVM         | --                    | --                  | --                    | --                  | 0.1741               | 0.3828              |
| LatentMLP   | 0.0435                | 0.8916              | 0.6951                | --                  | --                   | --                  |
| ConvLSTM    | 0.0256                | 0.9924              | 0.5433                | --                  | --                   | --                  |



We would like to thank once again all the Reviewers for their constructive remarks, comments and questions that greatly improved the quality of our work.

---

### Meta-Review · Area_Chair_MwCw · 2026-01-09

**Summary:**

The paper presents a new multimodal foundation model pre-trained on the recently-introduced BioCube dataset and targeted towards biodiversity-monitoring applications. Evaluations on two downstream tasks shows competitive performance (on those tasks) relative to other foundation models such as Aurora.

The paper received 4 comprehensive reviews, and the authors posted detailed responses to each reviewer. Concerns from the reviewers included scientific novelty (versus scientific integration), confusion around architectural choices (SwinT vs Perceiver), and confusion/lack of clarity around the particulars of the dataset.

But the primary concern from the reviewers --- which I echo fully after having read the manuscript --- pertained to the evaluation section. Only two downstream tasks were presented, and only one that is obviously biodiversity-centric. So the claim of it being a "foundation model for biodiversity" is over-stated in my opinion. Many more comparisons with both supervised and self-supervised baselines on a range of different tasks would have helped. The comparisons with Aurora (which is a climate model) seemed like apples-to-oranges, and the tasks themselves were somewhat under-specified.

I think this is a valuable direction for research and encourage the authors to carefully consider reviewer feedback as they proceed further.

**Reviewer Concerns:**

I think the scientific novelty question could have perhaps been resolved with some amount of back and forth. The clarity of writing also could have perhaps been resolved with discussion and an update to the manuscript. The fundamental flaw (the lack of comprehensive evals) would have been still outstanding.

**Reviewer Scores:**

There was a brief discussion among the reviewers before the discussion cutoff date which echoed earlier conerns, so I do not think any score would have changed.

---

### Decision · Program_Chairs · 2026-01-26

Reject